# QUANTUM ALGORITHM FOR DEEP NEURAL NETWORKS WITH EFFICIENT I/O

## ABSTRACT

A primary aim of research in quantum computing is the realization of quantum advantage within deep neural networks. However, it is hindered by known challenges in constructing deep architectures and the prohibitive overhead of quantum data I/O. We introduce a framework to overcome these barriers, designed to achieve an asymptotic speedup over the large input dimension of modern DNNs. This framework is based on the belief that a deep learning model can achieve similar performance when "rough" copies of the data are allowed, which is called the good-enough principle in this paper. Our framework enables the design of multi-layer Quantum ResNet and Transformer models by strategically breaking down the task into subroutines and assigning them to be executed by quantum linear algebra (QLA) or quantum arithmetic modules (QAM). This modularity is enabled by a novel data transfer protocol, Discrete Chebyshev Decomposition (DCD). Numerical validation reveals a pivotal insight: the measurement cost required to maintain a target accuracy scales sublinearly with the input dimension, verifying the good-enough principle. This sublinear scaling is key to preserving the quantum advantage, ensuring that I/O overhead does not nullify the computational gains. A rigorous resource analysis further corroborates the superiority of our models in both efficiency and flexibility. Our research provides strong evidence that quantum neural networks can be more scalable than classical counterparts on a fault-tolerant quantum computer.

## 1 INTRODUCTION

The current era of artificial intelligence is defined by the triumph of deep neural networks (DNNs). Large-scale models, particularly the Transformer architecture Vaswani et al. (2017), have revolutionized countless fields by leveraging immense depth to learn complex data representations. This success, however, is a double-edged sword. The computational demands of these models create a formidable bottleneck, especially for operations whose complexity scales polynomially with the primary input dimension, such as the $O(N^2)$ attention mechanism in Transformers with sequence length $N$. In parallel, quantum computing offers a new paradigm promising significant speedups for such tasks Nielsen & Chuang (2010); Preskill (2018). This has catalyzed research into Quantum Deep Neural Networks (QDNNs) Beer et al. (2020); Liu et al. (2024); Li et al. (2020b); Kerenidis et al. (2020); Ye et al. (2025), aiming to harness quantum mechanics to transcend the scaling limitations of classical deep learning.

However, the pursuit of practical QDNNs has splintered into two main directions, each with fundamental limitations. Variational Quantum Circuits (VQCs) Cerezo et al. (2021); Wen et al. (2024); Evans et al. (2024) are compatible with near-term hardware but generally lack provable speedups and are plagued by trainability issues like barren plateaus McClean et al. (2018); Wang et al. (2021); Anschuetz & Kiani (2022); Bittel & Kliesch (2021). Conversely, approaches based on Quantum Linear Algebra (QLA) subroutines Kerenidis & Prakash (2016); Childs et al. (2017); Liu et al. (2021); Krovi (2023); Liao & Ferrie (2024) promise demonstrable polynomial speedups. Yet, these QLA-based methods confront a critical challenge in constructing genuinely deep architectures, the quantum no-clone theory. Attempts have been made to circumvent this problem by leveraging iterative communication between classical and quantum systems Kerenidis et al. (2020). The quantum data I/O bottleneck, which theoretically bounds the overhead of faithfully reconstructing a quantum state, has largely confined these proposals to feasible constructs.

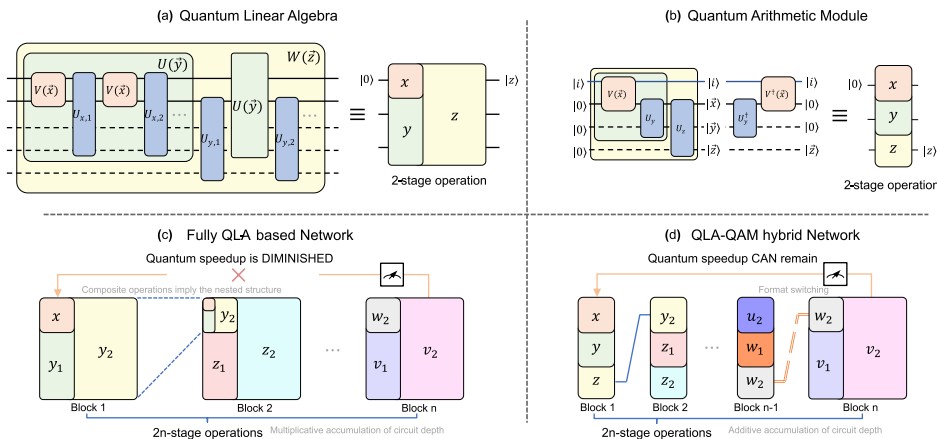

Figure 1. An overview of our hybrid quantum-classical framework for building deep quantum neural networks. (a)/(b) Typical examples of quantum linear algebra (QLA) and quantum arithmetic module (QAM) circuits and their symbolic representations. (c) The symbolic representation of existing quantum networks based on nested QLA operations, which suffer from multiplicative complexity scaling. (d) Our QLA-QAM Hybrid Network, which utilizes the additive circuit depth of QAMs to realize a practical quantum speedup.

In this paper, we address these fundamental limitations by proposing a novel quantum deep neural network framework. One core insight is to achieve practical quantum advantage by selectively accelerating only the computationally intensive parts of a DNN that bottleneck with respect to the large input dimension ($N$) Vaswani et al. (2017); Dao et al. (2022), while efficiently handling operations on the smaller, fixed feature dimension ($d$) with more flexible quantum routines. This targeted acceleration strategy leads to a hybrid quantum-classical, layer-by-layer execution model, whose design principles are illustrated in Figure 1. Our framework systematically decomposes DNN layers into three module types: Quantum Linear Algebra Modules (QLAs, green blocks) for operations that scale with the large dimension $N$, and Quantum Arithmetic Modules (QAMs, blue blocks) for efficient processing along the feature dimension $d$. The entire deep, modular architecture is made feasible by a "good-enough" information transfer principle: a deep network can achieve high performance without perfect, high-fidelity reconstruction of its intermediate states. This "good-enough" principle, inspired by the robustness of classical networks to quantization and pruning Han et al. (2015), posits that preserving salient features is more critical than exact state replication. Inspired by this observation, a novel protocol we term Discrete Chebyshev Decomposition (DCD) is proposed to improve the notorious quantum data I/O bottleneck between layers.

As depicted in Figure 1a-b, QLAs and QAMs exhibit fundamentally different scaling properties. A critical distinction lies in their composition: composing multiple QLA operations, as in existing proposals (Figure 1c), leads to a multiplicative accumulation of circuit complexity and error, often rendering theoretical speedups impractical for deep architectures. In sharp contrast, the circuit depth of our QAMs accumulates only additively. This linear scaling is crucial for constructing deep networks, enabling efficient element-wise non-linearities and parallel dot products.

This architectural choice directly addresses the viability of quantum speedup in deep networks. A fully QLA-based network (Figure 1c) struggles due to the compounding complexity of nested QLA subroutines. Our QLA-QAM hybrid model (Figure 1d), however, leverages the additive depth of QAMs to create a feasible pathway to acceleration, with the overall speedup ultimately depending on a manageable sampling cost. These modules can be flexibly assembled to form sophisticated architectures like a Quantum ResNet or a Quantum Transformer, demonstrating the framework's versatility. The DCD protocol then acts as the crucial bridge, enabling robust inter-layer communication without prohibitive overheads.

This work presents the first concrete theoretical and empirical validation of this targeted acceleration strategy as a viable pathway toward large-scale QDNNs. Our key contributions are summarized as follows:

**A Hybrid Quantum Acceleration Framework for Deep Networks:** We propose a novel framework that systematically decomposes deep neural network operations. It strategically allocates large-dimension computations to QLA and small-dimension, parallel tensor operations to efficient QAMs. This design enables the construction of multi-layer Quantum ResNet and Transformer models with a provable end-to-end speedup.

**Demonstrating and Exploring I/O Overhead for QDNNs:** We fully take the scaling of measurement cost for quantum deep learning models into consideration. Specifically, we investigate how the measurement cost, required to maintain target accuracy, scales with the input dimension. We also introduce the Discrete Chebyshev Decomposition (DCD) protocol, a novel and efficient "good-enough" data transfer mechanism for mitigating quantum I/O bottleneck, which demonstrates reduced dependence on system size.

**Resource Analysis and Practical Advantage:** Through detailed theoretical and numerical resource analysis, we quantitatively demonstrate that our hybrid approach significantly outperforms state-of-the-art fully quantum-based proposals. We further conduct a comprehensive assessment of the DCD protocol to identify the conditions under which it offers distinct advantages precisely and to quantify the extent of those benefits.

### 1.1 RELATED WORKS

**Quantum Neural Networks** Research on quantum neural networks has made significant progress Zhao & Wang (2021); Valdez & Melin (2023); Peral-García et al. (2024). Levine et al. (2019) has established theoretical connections between deep learning architectures and quantum entanglement. VQCs are one major direction for quantum neural networks Mitarai et al. (2018); Cong et al. (2019); Cerezo et al. (2021). They are compatible with near-term hardware but face severe trainability issues, such as barren plateausMcClean et al. (2018); Anschuetz & Kiani (2022). However, fault-tolerant algorithms based on quantum linear algebra (QLA) can offer provable speedups Kerenidis et al. (2020); Guo et al. (2024), while both data I/O and the scaling of network depth constitute significant hurdles. Beyond these categories, alternative learning paradigms have also been explored Amin et al. (2018); Pan et al. (2023); Ye et al. (2025).

**Measurement Techniques** The quantum-classical I/O has been mitigated by classical preprocessing in Stein et al. (2022); Kwak et al. (2023). Novel measurement techniques have been proposed, such as shadow tomography Aaronson (2018); Huang et al. (2020), which have applications in neural networks, as seen in Abbas et al. (2023).

**Quantum algorithms** Quantum Arithmetic algorithms, such as quantum adders and multipliers, have been optimized over the past decades Draper (2000); Gidney (2018; 2019). Varieties of quantum Linear Algebra (QLA) algorithms have been proposed, including Harrow et al. (2009); Childs et al. (2017); Gilyén et al. (2019), and applied in machine learning tasks Lloyd et al. (2014); Kerenidis & Prakash (2016).

## 2 QUANTUM MODULES

### 2.1 QUANTUM LINEAR ALGEBRA (QLA)

Quantum Linear Algebra is a type of quantum algorithm that has wide applications in the field of data analysis, including notable algorithms such as quantum component analysis Lloyd et al. (2014), quantum linear system solvers Harrow et al. (2009); Wossnig et al. (2018), and quantum differential equation system solvers Berry et al. (2017); Xue et al. (2021); Liu et al. (2021).

In QLA, a matrix $A$ is usually encoded into a quantum state by amplitude encoding Nakaji et al. (2022); Gonzalez-Conde et al. (2024):

$$|A\rangle = \frac{1}{\sqrt{\sum_{i,j} A_{ij}^2}} \sum_{i,j} A_{ij} |i\rangle |j\rangle, \tag{1}$$

or into quantum operations by block-encoding, for some constant $\alpha$ Wan et al. (2021):

$$(I \otimes \langle 0|)U_A(I \otimes |0\rangle) = \frac{A}{\alpha} \longleftrightarrow U_A = \begin{pmatrix} A/\alpha & \cdot \\ \cdot & \cdot \end{pmatrix}. \tag{2}$$

The matrix multiplication $AB$ is simple in QLA by $U_A |B\rangle$. The quantum singular value transformation even allows for a polynomial transformation $U_{p(A)}$ with polylogarithmic usage of $U_A$ Gilyén et al. (2019). Solving problems usually requires a composite of such operations. Due to the non-clone theorem in quantum mechanics, the quantum circuit behaves as a nested structure for successive transformation, as depicted in Figure 1a, of which the overhead will accumulate multiplicatively.

## 2.2 QUANTUM ARITHMETIC MODULES (QAMS)

Quantum Arithmetic Modules (QAMs) are the cornerstone for implementing operations on the smaller, fixed feature dimension ($d$) within our framework. The primary strength of QAMs lies in their ability to perform complex arithmetic operations in parallel. Given the input $|a\rangle, |b\rangle$, the function of a typical quantum adder can be written as Draper (2000); Ruiz-Perez & Garcia-Escartin (2017); Li et al. (2020a; 2021)

$$U_{\text{Add}} |a\rangle |b\rangle |0\rangle = |a\rangle |b\rangle |a + b\rangle. \tag{3}$$

The linearity and quantum superposition allow the parallel implementation:

$$U_{\text{Add}} \sum_i \alpha_i |i\rangle |a_i\rangle |b_i\rangle |0\rangle = \sum_i \alpha_i |i\rangle |a_i\rangle |b_i\rangle |a_i + b_i\rangle. \tag{4}$$

By combining quantum Adders and Multipliers, complex operations such as tensor products or contractions can be realized in parallel. Consider the operation $R_{ikjl} = \sum_\mu S_{i\mu j} T_{k\mu l}$ for tensors $S \in \mathbb{R}^{c_s \times d \times p}$ and $T \in \mathbb{R}^{c_t \times d \times q}$. The process, whose circuit structure is abstractly represented in Figure 1b, typically involves:

**State Preparation**: Input tensors $S$ and $T$ are loaded into quantum registers using controlled state preparation oracles, e.g., $O_S^{(c)} |i, j\rangle |0\rangle = |i, j\rangle \bigotimes_\mu |S_{i\mu j}\rangle$.

**Parallel Computation**: A series of quantum multiplier and adder circuits compute the products $S_{i\mu j} T_{k\mu l}$ for all $\mu$ in parallel and sum them into an accumulator register, resulting in the state $|i, j, k, l\rangle |R_{ikjl}\rangle$.

**Uncomputation**: To release ancillary qubits for reuse and maintain circuit reversibility, the inverse of the computation steps is applied to the input registers, returning them to their initial state $|0\rangle$, as visually suggested in Figure 1b.

This arithmetic-based approach allows for the efficient execution of structured linear algebra and element-wise non-linearities, providing the necessary computational primitives for deep learning layers while maintaining an additive, manageable growth in circuit complexity.

## 3 A FRAMEWORK FOR DEEP QUANTUM NETWORKS

### 3.1 INTRA-LAYER COMPUTATION: QLAS AND QAMS

Our framework's design is tailored for the common regime where a large input dimension $N$ dominates a smaller feature dimension $d$ (e.g., sequence length vs. embedding dimension in Transformers). Our strategy is to achieve quantum speedup specifically with respect to $N$. This dictates a modular separation of labor within each layer between two distinct types of quantum modules.

Figure 2 visually contrasts these two approaches.

**Quantum Linear Algebra Modules (QLAs)** are reserved for operations that are computationally dense and scale with the large dimension $N$, as shown in Figure 2a. They operate on amplitude-encoded data and utilize block-encoding algorithms to tackle the primary bottlenecks, such as the $N \times N$ matrix multiplications in Transformer attention.

Table 1: Complexity Comparison of Information Extraction Protocols for a state in $\mathbb{C}^d$.

| Protocol | Complexity | Classical Post-processing | Goal |
|---|---|---|---|
| **Full QST** | $O(d^2)$ | $O(d^3)$ | Full density matrix reconstruction |
| **Shadow Tomography**[a] | $O(K \log(M)/\delta^2)$ | $O(M \cdot \mathrm{poly}(\log d))$ | Estimate $M$ few-body observables |
| **DCD Protocol (Our work)** | $O(r/\delta)$ | $O(r \cdot d)$ | **Extract $r$ global feature coefficients** |

[a]For estimating $M$ observables with Pauli weight at most $K$ to precision $\delta$.

**Quantum Arithmetic Modules (QAMs)** handle computations that are sparse, element-wise, or structured along the smaller dimension $d$. As illustrated for the ReLU activation in Figure 2b, QAMs operate on digitally encoded numbers. They are essential for applying non-linearities (e.g., ReLU) and performing structured linear algebra where operations can be parallelized over the $N$ items (e.g., applying a $d \times d$ weight matrix to $N$ vectors).

The synergy between QLAs and QAMs is the key to handling modern deep learning models: QLAs provide the speedup for large-scale, dense linear algebra, while QAMs efficiently implement the necessary non-linear activations and feature-space transformations. This combination allows for a faithful and accelerated quantum implementation of entire network layers.

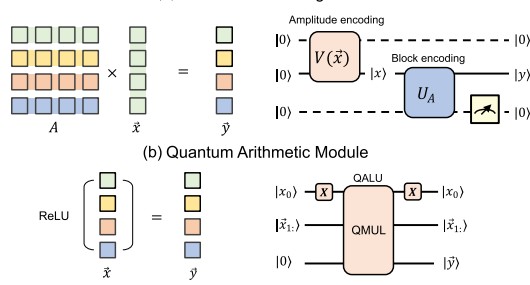

Figure 2. Mapping dense and sparse operations to corresponding quantum modules. (a) The dense lattice diagram of matrix multiplication is implemented with a QLA module. (b) The sparse, element-wise nature of the ReLU function is implemented with a QAM.

### 3.2 Inter-Layer Communication: The Discrete Chebyshev Decomposition Protocol

The critical link in our architecture—and the concrete embodiment of our "good-enough" principle—is the protocol for information conversion between classical and quantum computer. To bypass the infeasible cost of tomography, we introduce the Discrete Chebyshev Decomposition (DCD) protocol, designed to extract a compressed classical representation of a quantum state.

Our choice of the Chebyshev basis is mathematically motivated. For any function on a finite interval, a truncated Chebyshev series provides the best polynomial approximation in the $l_\infty$ norm (minimax approximation) Ahmed et al. (2006); Trefethen (2019). Chebyshev basis also has natural applications in QLA algorithms Martyn et al. (2021). We view the amplitudes of a quantum state $|\psi\rangle$ as evaluations of an underlying function. By projecting $|\psi\rangle$ onto the first $r$ Chebyshev basis vectors, we find the optimal low-degree polynomial approximation of this function, capturing its most significant, low-frequency features with a minimal number of coefficients.

The DCD protocol assumes that the information in a layer's output state $|\psi\rangle$ is highly compressible. Any such state can be formally expanded in the discrete Chebyshev basis $\{|T_j\rangle\}$ as $|\psi\rangle = \sum_{j=0}^{d-1} c_j |T_j\rangle$, where $c_j = \langle T_j|\psi\rangle$. Our core hypothesis is that an approximation using only the first $r \ll d$ coefficients is sufficient for the next layer. The protocol is detailed in Algorithm 1.

**Theorem 3.1.** *(Discrete Chebyshev Decomposition) Given access to a state preparation unitary for $|\psi\rangle$ with cost $C_\psi$, the DCD protocol can estimate the first $r$ Chebyshev coefficients to precision $\delta$ with total query complexity $\tilde{O}(r \cdot C_\psi/\delta)$. The subsequent state re-preparation for the next layer requires $O(r)$ digital encoding input and a QAM with complexity $\tilde{O}(r \cdot poly(\log d))$.*

The efficiency of the DCD protocol is its main advantage. As summarized in Table 1, DCD offers a clear advantage over the intractable scaling of QST. While shadow tomography is effective for estimating local observables, DCD is purpose-built to extract a global, spectral representation of the state. Its query complexity scales only with the desired number of features, $r$, and precision, $\delta$.

**Algorithm 1:** Discrete Chebyshev Decomposition (DCD) Protocol

**Input:** Output state of layer $k$, $|\psi_{\text{out}}^{(k)}\rangle$; truncation rank $r$; target precision $\delta$.
**Output:** Classical coefficient vector $\mathbf{c}_{\text{classical}} = [c_0, c_1, \ldots, c_{r-1}]^T$.

/* Coefficient Estimation                                                    */
1 **for** $j \leftarrow 0$ **to** $r - 1$ **do**
2      Efficiently prepare the basis state $|T_j\rangle$ using its known recurrence relation;
3      Construct a circuit to project $|\psi_{\text{out}}^{(k)}\rangle$ onto $|T_j\rangle$;
4      Use Quantum Amplitude Estimation (QAE) to estimate the coefficient $c_j = \langle T_j | \psi_{\text{out}}^{(k)}\rangle$ to
     precision $\delta$;
5      Store the estimated real value $c_j$ classically;
6 **end**

/* State Re-preparation                                                      */
7 Load the classical vector $\mathbf{c}_{\text{classical}}$ into quantum digital encoding;
8 Use a QAM to compute the amplitudes of the approximate vector $\tilde{\psi}_i = \sum_{j=0}^{r-1} c_j T_{ji}$ for each
     computational basis state $|i\rangle$;
9 Prepare the input state for layer $k + 1$, $|\psi_{\text{in}}^{(k+1)}\rangle = \sum_i \tilde{\psi}_i |i\rangle$, using a standard state preparation
     routine.

Since our work demonstrates that $r$ can be significantly smaller than $d$, DCD transforms data transfer from an insurmountable bottleneck into a manageable subroutine, making an end-to-end quantum speedup for deep learning finally achievable.

### 3.3 QUANTUM MODEL INSTANCES: qRESNET & qTRANSFORMER

To demonstrate the versatility of our framework, we now instantiate it by constructing a quantum Residual Network (qResNet) and a quantum Transformer (qTransformer). These examples showcase how our modular approach maps classical computational patterns onto the most suitable quantum primitives, guided by the principle of targeting speedups relative to the primary input dimension (e.g., sequence length $N$ or image size $H \times W$). The concrete quantum implementations of them can be found in Appendix B.2 and B.3, together with the proof of corresponding theorems.

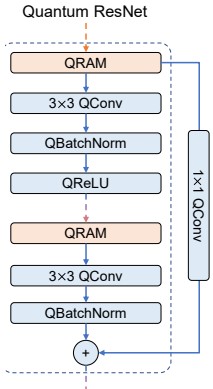

Quantum ResNet

Figure 3. The quantum realization of ResNet.

**Quantum ResNet (qResNet)** Our Quantum ResNet (qResNet) adapts the architecture of a classical ResNet-18 He et al. (2016), as depicted in Figure 3. We focus on the regime where the image's spatial dimensions $(H, W)$ are significantly larger than the channel dimension $(C)$. The core computational tasks are Convolution, Activation, and Residual Connections, which are implemented using the QAM A key design choice is the use of our Data Transfer Module (DTM) after each QAM-based layer. This prevents the composition of multiple sparse operators from creating a dense, computationally complex transformation, thereby preserving the efficiency of the QAM throughout the network's depth. The complexity is summarized in Theorem 3.2.

**Theorem 3.2.** *(Quantum ResNet Block) Given a ResNet Block, whose input tensor and kernel have shapes of $(B, C, H, W)$ and $(C, C, K, K)$ respectively, a quantum implementation of the ResNet Block has the quantum overhead of $\tilde{O}(CK^2 \times S(B, C, H, W))$, where $S(B, C, H, W)$ is the sampling overhead of a quantum state with shape of $(B, C, H, W)$. The number of queries to the input data and kernel is twice for each implementation.*

**Quantum Transformer (qTransformer)** For vision tasks, we implement an encoder-only Quantum Transformer, focusing on the common $N \gg d$ regime, where $N$ is the sequence length and $d$ is the embedding dimension. This assumption dictates the allocation of tasks to create a highly efficient quantum analog, where the Feed-Forward Net-

work and Residuals are implemented solely by QAM. Multi-Head Self-Attention(MHSA) has a hybrid implementation of QLA and QAM. This modular design is shown in Figure 4, where the blue, green, or orange blocks represent QAM, QLA, or DTM, respectively.

The complexity of a quantum Encoder Block is given in Theorem 3.3

**Theorem 3.3.** *(Quantum Encoder Block) For an input tensor with shape of $(B, N, d)$, where $N$ is the number of tokens and $d$ is the token length, the Quantum overhead of the Quantum Encoder Block is $\tilde{O}(d^2 \times S(B, N, d))$. The number of queries to the input data $X$ is 6 for each implementation.*

The flowchart of the MHSA module (Figure 4b) showcases this strategic division of labor. This deliberate allocation of computational tasks—reserving the QLA for the true $N$-dimensional bottlenecks and using the QAM for structured, $d$-dimensional arithmetic—is the key to achieving a significant asymptotic speedup with respect to the input sequence length.

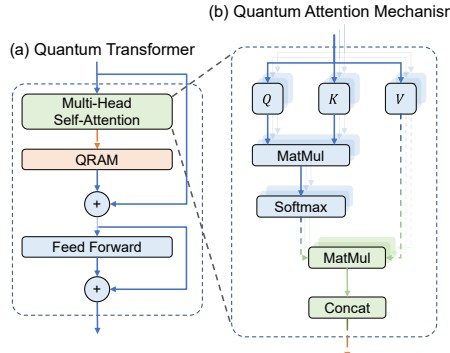

Figure 4. The quantum realization of ResNet. The red, blue, and green blocks represent DTM, QAM, and QLA, respectively

# 4 EXPERIMENTS

This section presents numerical experiments designed to validate the efficacy and advantages of our proposed quantum-classical hybrid framework. We begin by demonstrating the superior efficiency and favorable scaling properties of our Discrete Chebyshev Decomposition (DCD) protocol when contrasted with a standard $l_\infty$-norm tomography baseline Kerenidis et al. (2020). Subsequently, we delve into a detailed resource analysis of Quantum ResNet (qResNet) and Quantum Transformer (qTransformer) architectures, aiming to quantitatively establish the practical benefits of our integrated approach on well-established image classification benchmarks. In our experiments, the datasets we mainly use are CUB-200-2011 Wah et al. (2011), where the input dimension $N$ represents the sequence length for Transformers and the spatial pixel count $H \times W$ for ResNet.

In our experiments, the quantum models are simulated on classical hardware. During the training phase, we implement a hybrid procedure: the forward pass simulates the quantum inference process, explicitly incorporating the sampling noise introduced by different measurement protocols to ensure the model adapts to the approximation errors. Parameter optimization (backward pass), however, is computed via standard classical backpropagation. The reported performance metrics are then evaluated using this quantum-simulated inference on the test set.

## 4.1 EMPIRICAL VALIDATION OF DCD EFFICIENCY AND SCALING

Figure 5 provides empirical validation of our DCD protocol's performance against the $l_\infty$ method for both qResNet (top row, **a-c**) and qTransformer (bottom row, **d-f**). For each data transfer method, our initial step involves identifying the minimum hyperparameter configuration (specifically, sampling precision for $l_\infty$ and rank $r$ for DCD) necessary to achieve at least 95% of the classical model's peak performance (Figure 5a, b, d, e). Following this, we illustrate the total quantum overhead $Q$ associated with these optimized settings as a function of the input dimension $N$ (Figure 5c, f).

For qResNet, while both data transfer methods successfully attain the predefined target accuracy, their associated resource costs exhibit a significant divergence. As clearly

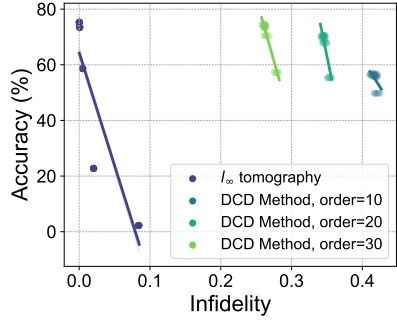

Figure 6. The relationship between classification accuracy and the infidelity of quantum state re-preparation.

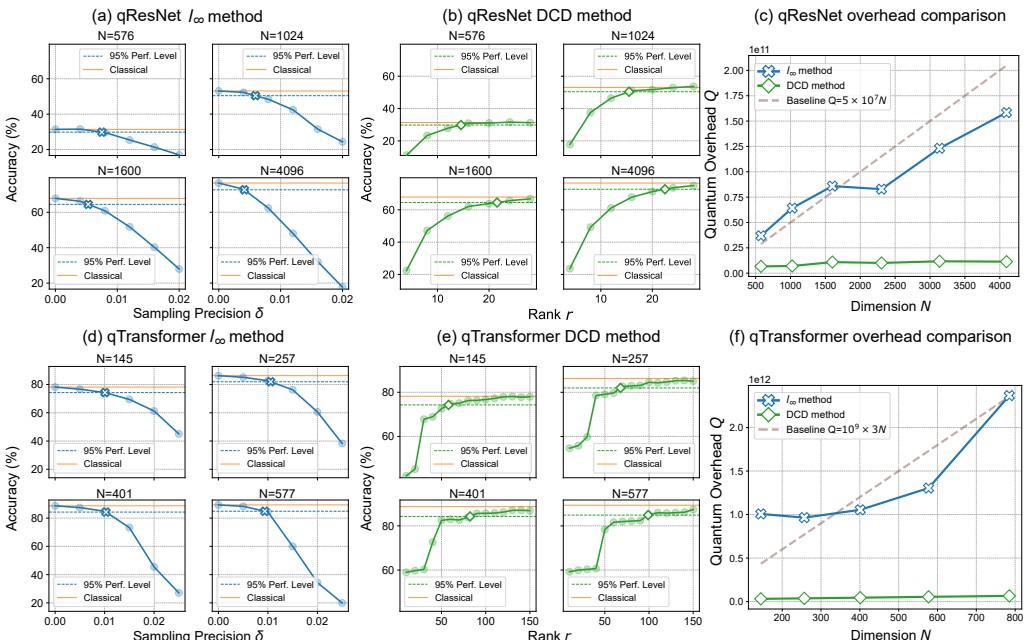

Figure 5. (a)/(d) Classification accuracy of the $l_\infty$ tomography method for quantum ResNet-18 and Transformer across varying input dimensions $N$. The blue dashed line indicates the 95% performance threshold relative to the classical model, highlighted by the orange line. (b)/(e) Classification accuracy of the DCD method for both models. (c)/(f) Relationship between computed quantum resources and corresponding input dimensions $N$ for both methods, with a linear baseline plotted for comparison.

depicted in Figure 5c, the quantum overhead for the $l_\infty$ method scales approximately linearly with the input dimension $N$. The DCD overhead remains low, demonstrating its superior scaling properties for qResNet.

This inherent advantage of DCD becomes even more pronounced when applied to the qTransformer architecture. The DCD rank analysis, presented in Figure 5e, reveals a distinct "elbow" effect, characterized by a sharp initial increase in accuracy followed by a clear plateau. This observation strongly suggests that DCD is highly effective at identifying and leveraging a compact, yet information-rich, subspace within the qTransformer's learned representations. This intrinsic efficiency directly translates into a dramatic reduction in quantum overhead, with DCD's computational cost scaling sublinearly with $N$ (Figure 5f). This empirically observed sublinear scaling of measurement cost stands as a central and pivotal result of our study, providing compelling evidence that our DCD protocol significantly enhances the scalability of quantum deep learning by effectively mitigating the notorious I/O bottleneck.

## 4.2 ANALYSIS OF QUANTUM PROPERTIES AND HYBRIDIZATION

To gain deeper insights into how intrinsic quantum properties and architectural choices influence the performance of our quantum deep learning models, particularly qResNet and qTransformer, we conducted two key analyses.

First, we thoroughly investigated the impact of quantum state re-preparation infidelity on model performance across different Data Transfer Method (DTM) protocols. As illustrated in Figure 6, the DCD method consistently achieves comparable classification accuracy even with significantly lower state infidelity compared to the $l_\infty$ method. This compelling observation robustly validates the 'good-enough' principle in the context of quantum deep learning. It suggests that a degree of redundancy inherently exists in the data transmission pathways of deep neural networks, opening

up substantial potential for quantum models to demonstrate performance advantages even under imperfect state preparation, by efficiently capturing the most salient features.

Second, we explored the performance evolution of our quantum Transformer model under varying degrees of quantum influence. This was achieved by gradually increasing the "quantumness" of the model, specifically by progressively replacing classical layers with their quantum counterparts. Figure 7 strikingly demonstrates that this "semi-quantum" or hybrid model effectively preserves a substantial portion of its performance, particularly during the initial stages of "quantization." This robustness is especially evident when employing the DCD method for data transfer, highlighting its resilience to partial quantum integration and its potential for practical, near-term hybrid implementations. This analysis underscores the flexibility and potential for incremental adoption of quantum components within classical architectures.

### 4.3 COMPREHENSIVE NUMERICAL RESOURCE ANALYSIS

To further quantitatively substantiate the advantages of our proposed framework, we present a detailed and concrete resource analysis. Table 2a provides a direct comparison between our quantum-classical hybrid framework and a prior, fully Quantum Linear Algebra (QLA)-based model Kerenidis et al. (2020), as well as an intra-framework comparison between the $l_\infty$ tomography and our DCD protocol. For clarity, we highlight in bold the outcomes that signify the most efficient resource utilization while maintaining equivalent or even marginally superior performance. The results demonstrate that our hybrid model achieves notably superior performance at a significantly reduced computational cost compared to the fully QLA-based model, thereby strongly confirming the inherent efficiency and architectural advantages of our design.

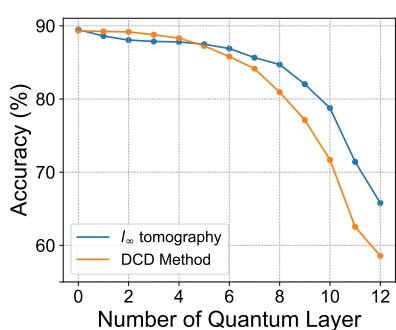

Figure 7. The relationship between model performance and the number of quantum layers within the hybrid Transformer architecture.

Furthermore, within Table 2, we conduct a thorough analysis of the critical trade-off between the two distinct data transfer methods employed within our framework. We compare the classification accuracy and the associated quantum resource cost for both qResNet and qTransformer when utilizing either the $l_\infty$-tomography or our proposed DCD protocol. The empirical findings robustly indicate that for most of the given range of desired performance, the DCD protocol consistently offers a substantial and compelling resource advantage. While $l_\infty$-tomography might achieve a marginally higher peak accuracy, it invariably incurs this at a disproportionately greater quantum cost. This stark contrast emphatically underscores DCD's superior performance-to-cost ratio, making it an exceptionally attractive choice, especially in scenarios where quantum resources are inherently constrained. It also suggests that while DCD offers significant gains, these returns may diminish as one pushes towards the absolute theoretical performance limits of the model. Collectively, these comprehensive results provide direct and robust numerical evidence for the practical efficiency, scalability, and overall efficacy of our integrated quantum-classical hybrid framework.

## 5 DISCUSSION

We presented a hybrid quantum-classical framework tackling the scalability challenges in quantum deep learning. Central to this design is the Discrete Chebyshev Decomposition (DCD) protocol, which alleviates the quantum I/O bottleneck and enables favorable scaling in resource overhead while maintaining improved fidelity. Resource analysis indicates that under efficient I/O mitigation, quantum advantage in deep neural networks is within reach, paving the way for more capable architectures.

This framework benefits from the compressibility of intermediate states, in line with spectral bias in classical deep learning Rahaman et al. (2019). However, its effectiveness may diminish in domains dominated by high-entropy or high-frequency features, requiring larger truncation ranks and

Table 2: Comparison of DCD and $l_\infty$ Tomography for quantum ResNet and Transformer.

(a) Quantum ResNet

| Model | Rank | Sampling Precision | | | | | | | | | |
|---|---|---|---|---|---|---|---|---|---|---|---|
| | | 0.002 | | 0.004 | | 0.010 | | 0.020 | | 0.040 | |
| | | Accuracy ↑ (%) | Overhead ↓ ($\times10^9$) | Accuracy ↑ (%) | Overhead ↓ ($\times10^9$) | Accuracy ↑ (%) | Overhead ↓ ($\times10^9$) | Accuracy ↑ (%) | Overhead ↓ ($\times10^9$) | Accuracy ↑ (%) | Overhead ↓ ($\times10^9$) |
| **DCD** | 10 | 56.44 | 20.44 | 56.80 | 10.23 | 56.23 | 4.10 | 55.68 | 2.06 | 49.78 | 1.03 |
| | 20 | 70.33 | 81.76 | 70.54 | 40.91 | 69.30 | **16.39** | 67.85 | **8.22** | 55.35 | 4.14 |
| | 30 | 74.16 | 183.95 | 74.49 | **92.04** | 73.32 | **36.87** | 70.47 | **18.50** | 57.21 | 9.31 |
| $l_\infty$ Tomo. | – | 75.27 | **989.56** | 73.47 | 247.39 | 58.65 | 39.58 | 22.76 | 9.90 | 2.24 | 2.47 |
| | $M$ | Accuracy ↑ (%) | Overhead ↓ ($\times10^{15}$) | Accuracy ↑ (%) | Overhead ↓ ($\times10^{15}$) | Accuracy ↑ (%) | Overhead ↓ ($\times10^{15}$) | Accuracy ↑ (%) | Overhead ↓ ($\times10^{15}$) | Accuracy ↑ (%) | Overhead ↓ ($\times10^{15}$) |
| **QLA Model** | $10^3$ | 69.23 | 2.87 | 66.53 | 0.72 | 52.64 | 0.15 | 20.19 | 0.03 | 1.74 | 0.01 |
| | $10^5$ | 74.77 | 287.24 | 73.44 | 71.81 | 59.39 | 11.49 | 22.99 | 2.87 | 1.47 | 0.72 |

(b) Quantum Transformer

| Model | Rank | Sampling Precision | | | | | | | | | |
|---|---|---|---|---|---|---|---|---|---|---|---|
| | | 0.0002 | | 0.0004 | | 0.0010 | | 0.0020 | | 0.0040 | |
| | | Accuracy ↑ (%) | Overhead ↓ ($\times10^{11}$) | Accuracy ↑ (%) | Overhead ↓ ($\times10^{11}$) | Accuracy ↑ (%) | Overhead ↓ ($\times10^{11}$) | Accuracy ↑ (%) | Overhead ↓ ($\times10^{11}$) | Accuracy ↑ (%) | Overhead ↓ ($\times10^{11}$) |
| **DCD** | 40 | 59.23 | 5.25 | 58.34 | 2.54 | 52.35 | 0.95 | 44.60 | 0.46 | 31.12 | 0.22 |
| | 60 | 81.76 | 7.88 | 81.91 | 3.81 | 80.32 | 1.42 | 76.01 | **0.69** | 57.51 | 0.33 |
| | 150 | 86.81 | **19.69** | 86.78 | **9.53** | 86.50 | **3.56** | 84.86 | **1.72** | 78.05 | **0.83** |
| | | 0.0050 | | 0.0100 | | 0.0150 | | 0.0200 | | 0.0250 | |
| $l_\infty$ Tomo | – | 88.38 | **45.82** | 84.38 | 11.45 | 60.03 | 5.09 | 34.36 | 2.86 | 19.92 | 1.83 |

resulting in reduced asymptotic speedups. Quantum-inspired classical baselines remain valuable for benchmarking, though their precision scaling is less favorable for deep architectures compared to our complexity Tang (2019); Arrazola et al. (2019); Tang (2021); Chia et al. (2022).

Future work will focus on: identifying optimal I/O protocols beyond DCD, rigorously characterizing the applicability boundaries of the low-rank assumption, and exploring model design strategies that jointly achieve quantum acceleration and maintain—or surpass—classical performance while meeting re-preparation fidelity requirements.

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

# A  PRELIMINARIES

To construct our framework for deep quantum networks, we leverage advanced algorithms from quantum linear algebra and quantum arithmetic, applying them to emulate classical architectures like ResNet and the Transformer. This section aims to review these fundamental building blocks.

## A.1  QUANTUM SUBROUTINES FOR LINEAR ALGEBRA

Quantum Linear Algebra (QLA) promises significant speedups for classically intractable tasks, forming the computational core of many quantum machine learning proposals. While early algorithms like HHL demonstrated the potential for exponential advantage, modern QLA has largely converged around more versatile and robust techniques.

A central concept in modern QLA is block-encoding, a method for embedding a non-unitary matrix $A$ into a larger unitary matrix $U_A$. Specifically, an $(\alpha, a, \delta)$-block-encoding of $A$ is a unitary $U_A$ such that

$$(\langle 0|^{\otimes a} \otimes I)U_A(|0\rangle^{\otimes a} \otimes I) = A/\alpha, \qquad (5)$$

where $\alpha \geq \|A\|$ is a normalization factor, $a$ is the number of ancillary qubits, and the approximation is up to an error $\delta$. This technique transforms the problem of applying a matrix into the problem of implementing a unitary circuit, making it amenable to quantum computation. Many efficient block-encoding methods exist for structured matrices, such as sparse or low-rank matrices.

Once a matrix is block-encoded, its properties can be manipulated. For instance, the Quantum Singular Value Transformation (QSVT) Gilyén et al. (2019) provides a unified framework for applying polynomial functions of a matrix's singular values to a quantum state. While QSVT is a powerful and general tool, many QLA tasks, including those in our work, can be realized using a more fundamental subroutine: Quantum Amplitude Estimation (QAE) Brassard et al. (2002). QAE allows for the estimation of the amplitude of a specific basis state in a quantum superposition. For example, if a quantum state $|\psi\rangle$ is prepared such that the probability of measuring a target state $|0\rangle$ is $p = |\langle 0|\psi\rangle|^2$, QAE can estimate $p$ with an error $\delta$ using $O(1/\delta)$ queries to the state preparation circuit, achieving a quadratic speedup over classical sampling. This subroutine is crucial for extracting information from a quantum system, such as computing the inner product between two states or the expected value of an observable.

## A.2  QUANTUM ARITHMETIC FOR SPARSE AND ELEMENT-WISE OPERATIONS

While QLA excels at large-scale, dense matrix operations, DNNs also rely heavily on element-wise operations, such as adding biases, applying activation functions, and executing sparse transformations. These tasks necessitate Quantum Arithmetic (QA), which performs computations directly on the numerical values encoded in quantum registers, typically using a fixed-point binary representation.

QA circuits for fundamental operations like addition and multiplication have been well-established Vedral et al. (1996); Draper (2000), with resource costs (e.g., gate count and circuit depth) scaling polynomially with the precision (number of bits) of the encoded numbers. Furthermore, by adapting logic from classical circuits, quantum computers can efficiently perform general-purpose arithmetic operations.

**Corollary A.1.** *Given a function $f(x) : \mathbb{R} \rightarrow \mathbb{R}$ that can be efficiently estimated classically, there exists a quantum algorithm to implement quantum arithmetic $|x\rangle|0\rangle \rightarrow |x\rangle|\tilde{f}(x)\rangle$, where $|\tilde{f}(x) - f(x)| \leq \delta$ and $\delta$ represents the computing accuracy. The gate complexity of the algorithm is $O(\mathrm{polylog}(1/\delta))$.*

*Proof.* As discussed in Nielsen & Chuang (2010), quantum computing can efficiently simulate classical logic circuits using reversible gates. Therefore, any function efficiently computable classically is also efficiently computable on a quantum computer. $\square$

To exemplify the implementation of tensor operations, we assume the typical data $X \in \mathbb{R}^{a \times s}$ is digitally encoded by a quantum circuit $O_X^{(c)}$, whose output is the direct product of quantum bit strings of each component:

$$O_X^{(c)} |i\rangle |0\rangle = |i\rangle \bigotimes_j |X_{ij}\rangle . \tag{6}$$

A matrix-vector multiplication can be realized directly between $X$ and $v \in \mathbb{R}^s$ with this encoding and quantum arithmetic. Using quantum adders and multipliers, we can achieve step by step:

$$
\begin{aligned}
|i\rangle |0\rangle |0\rangle |0\rangle &\xrightarrow{\text{Data Loading}} |i\rangle \bigotimes_j |X_{ij}\rangle \bigotimes_k |v_k\rangle |0\rangle \\
&\xrightarrow{\text{Element wise multiplication}} |i\rangle \left( \bigotimes_j |X_{ij}\rangle |v_j\rangle \right) |0\rangle \\
&\xrightarrow{\text{Quantum adder}} |i\rangle \left( \bigotimes_{j \neq 0} |X_{ij}\rangle |v_j\rangle \right) |X_{i0}, v_0\rangle |0 + X_{i0}v_0\rangle \\
&\xrightarrow{\text{Step-by-step addition}} |i\rangle \left( \bigotimes_{j \neq 1} |X_{ij}\rangle |v_j\rangle \right) |X_{i1}, v_1\rangle |0 + X_{i0}v_0 + X_{i1}v_1\rangle \\
&\xrightarrow{s \text{ steps}} |i\rangle \left( \bigotimes_j |X_{ij}\rangle |v_j\rangle \right) |\sum_k X_{ik}v_k\rangle .
\end{aligned} \tag{7}
$$

The overall gate count, regardless of the data loading subprocedure, is then $O(s)$, which is independent of $a$. Analysis of other sparse operations is similar. While this method is slower than the highly parallelized approach of QLA for dense matrices, its complexity scales with the number of involved elements, making it an efficient choice for sparse problems. Furthermore, QA is the primary tool for implementing non-linear activation functions, typically by computing a piecewise polynomial approximation of the target function (e.g., ReLU), which involves a sequence of arithmetic comparisons and calculations similar to classical implementations.

### A.3 CLASSICAL ARCHITECTURES OF INTEREST

Our work focuses on developing quantum counterparts for two of the most influential DNN architectures.

The Transformer Vaswani et al. (2017) has become the de facto standard for sequence modeling tasks. Its core innovation is the self-attention mechanism, defined as $\text{Attention}(Q, K, V) = \text{softmax}(\frac{QK^T}{\sqrt{d_k}})V$. The primary computational bottleneck is the matrix multiplication $QK^T$, which scales as $O(N^2)$ with the sequence length $N$, making it a prime target for quantum acceleration via QLA.

The Residual Network (ResNet) He et al. (2016) introduced the concept of residual connections, $y = \mathcal{F}(x) + x$, where $\mathcal{F}(x)$ is a block of layers. This "shortcut" structure effectively mitigates the vanishing gradient problem, enabling the training of networks with hundreds or even thousands of layers. Quantum analogues of ResNet provide an ideal testbed for assessing the ability of a quantum framework to handle truly deep architectures.

### A.4 QUANTUM RANDOM ACCESS MEMORY AND RESOURCE TRADE-OFFS

The practical implementation of the quantum input model relies on Quantum Random Access Memory (QRAM) to load classical data vectors into quantum states. Specifically, for a classical dataset $B = \{b_j\}_{j=0}^{N-1}$, the QRAM operation performs the mapping:

$$\sum_{j=0}^{N-1} \alpha_j |j\rangle |0\rangle \xrightarrow{\text{QRAM}} \sum_{j=0}^{N-1} \alpha_j |j\rangle |b_j\rangle, \tag{8}$$

where $|j\rangle$ is the address register and $|b_j\rangle$ is the data register.

A detailed resource analysis by Clader et al. (2022) highlights that the cost of QRAM is not negligible and presents distinct trade-offs between circuit depth (runtime) and gate count (hardware size). They analyze two primary architectures: the Bucket-Brigade (BB) model, which is optimized for noise resilience, and the Select-Swap (SS) model, which offers tunable resource scaling.

We focus on the Select-Swap (SS) model as it allows for a flexible exchange between T-depth and T-count via a parameter $\lambda \in \{0, \ldots, \log N\}$. For a dataset of size $N = 2^n$:

- **Minimal T-Count Configuration ($\lambda \approx 0$):** This configuration minimizes the total number of physical resources. It achieves a T-gate count of $\mathcal{O}(N)$ and uses $\mathcal{O}(N)$ ancilla qubits. However, the circuit depth scales linearly as $\mathcal{O}(N)$, which may mitigate quantum speedups in time-critical applications.
- **Minimal T-Depth Configuration ($\lambda \approx n$):** To preserve the exponential or polynomial speedup of quantum algorithms, one typically prioritizes circuit depth. In this regime, the SS model achieves a T-depth of $\mathcal{O}(\mathrm{polylog}(N))$ (specifically $\mathcal{O}(n)$). The trade-off is a significant increase in spatial overhead, requiring $\mathcal{O}(N^2)$ T-count and ancilla qubits.

In our resource estimation, we assume the availability of QRAM optimized for T-depth (the second configuration) to ensure the overall algorithmic time complexity remains logarithmic with respect to the input dimension $N$. While this implies a hardware cost scaling polynomially with $N$, the query depth remains $\mathcal{O}(\mathrm{polylog}(N))$, consistent with the requirements for maintaining the asymptotic quantum advantage claimed in our framework.

## B IMPLEMENTATION DETAILS

### B.1 DCD PROTOCOL

The implementation of DCD protocol to measure a quantum state is based upon the Quantum Discrete Chebyshev Transformation (QDCT), the function of which can be written as

$$U_{DCT}|i\rangle = |T_i\rangle = \sum_j T_i(x_j)|j\rangle. \tag{9}$$

QDCT can be realized with elementary gates and the Quantum Fourier Transformation circuits as shown in Klappenecker & Rotteler (2001), with quantum overhead scaling logarithmic with the system size. The DCD protocol is to obtain the coefficients of each Chebyshev basis by quantum amplitude estimation, the complexity of which comes up naturally now:

*proof of Theorem 3.1.* The coefficients estimation stage includes 3 parts: state preparation, QDCT, and amplitude estimation. The complexity of state preparation and QDCT is $O(\mathrm{polylog}\, d \times C_\psi)$. The amplitude estimation will multiply the complexity by a factor $O(\frac{1}{\delta})$. There are $r$ coefficients required to be estimated. Therefore, the overall complexity is $O(r \cdot C_\psi/\delta)$. The coefficients loading and state computation compose the state re-preparation stage. The cost of them is linear with $r$, which gives the claimed re-preparation overhead. $\qquad \square$

### B.2 RESIDUAL LAYER

To construct deep quantum neural networks, we introduce a quantum analogue of the classical residual block, inspired by ResNet architectures. This block enables the training of deeper models by using shortcut connections to mitigate vanishing gradient problems. A single block operates on a quantum state encoding a feature map and is composed of a main path and a shortcut path. The data flow within the block is managed by a Data Transfer Module (DTM), which handles state preparation from classical data via QRAM and measurement for intermediate classical processing.

A typical quantum residual block executes the following sequence:

1.**Main Path:** The input state $|\psi_{\mathrm{in}}\rangle$, encoding the feature map $X$, is processed sequentially by a quantum convolutional layer ($U_{\mathrm{qConv}}$), a quantum batch normalization layer ($U_{\mathrm{qBN}}$), and a quantum ReLU activation ($U_{\mathrm{QReLU}}$). This sequence may be repeated, as in standard ResNet blocks.

2.**Shortcut Path:** The original input state $|\psi_{\mathrm{in}}\rangle$ is preserved.

3.**Addition & Final Activation:** The output state from the main path, $|\psi_{\mathrm{main}}\rangle$, is added to the shortcut state $|\psi_{\mathrm{in}}\rangle$ using a quantum arithmetic adder from the QAM. A final ReLU activation, $U_{\mathrm{QReLU}}$, is applied to the resulting state to produce the block's output state, $|\psi_{\mathrm{out}}\rangle$.

**Quantum Convolutional Layer (qConv).**    The qConv layer performs convolution using the Quantum Arithmetic Module (QAM). Its goal is to transform an input feature map state $|\psi_X\rangle$ into an output state $|\psi_Y\rangle$ where $Y$ is the convolution of $X$ with a classically-defined kernel $K$ satisfying

$$Y_{ijc} = (\mathrm{bias})_c + \sum_{c',\Delta i,\Delta j} K_{c,c',\Delta i,\Delta j} \cdot X_{i+\Delta i, j+\Delta j, c'}. \tag{10}$$

The operation can be described as follows: for each output pixel position $(i, j, c)$, the QAM applies a unitary $U_{\mathrm{qConv}}$ that computes the dot product arithmetically:

$$U_{\mathrm{qConv}} : |i, j, c\rangle |0\rangle \rightarrow |i, j, c\rangle |Y_{ijc}\rangle. \tag{11}$$

This computation leverages the quantum adders and multipliers within the QAM to perform the operation in superposition across all output positions. The complexity comes as follows:

**Lemma B.1** (Convolutional Layer). *Given the shape of the input tensor $X$ be $(B, C, H, W)$ together with the kernel shape $(C, C, K, K)$, the gate count of $U_{qConv}$ is $O(CK^2)$, while the gate depth is $O(\log(CK))$.*

*Proof.* In the proof, we disregard the number of bits for the data, as its generalization to more bits of floating-point numbers is straightforward. First, we use the quantum input model for classical data loading with data replication to prepare

$$|i, j, c\rangle |0^{\otimes c+2k}\rangle \rightarrow |i, j, c\rangle \bigotimes_{\Delta i, \Delta j, c'} |X_{i+\Delta i, j+\Delta j, c'}\rangle, \tag{12}$$

where $c = \lceil \log_2 C \rceil, k = \lceil \log_2 K \rceil$.The data replication process only increases the word size for data loading, which contributes polylogarithmically to the complexity. The data loading is similar for

$$|c\rangle |0\rangle \rightarrow |c\rangle \bigotimes_{\Delta i, \Delta j, c'} |K_{\Delta i, \Delta j, c, c'}\rangle. \tag{13}$$

Set $U_{\mathrm{qConv}}$ to be the inner product circuit on the last two registers, and the complexity is then $O(CK^2)$, as discussed in Section A.2. We achieve the claimed operation and the claimed complexity. The gate depth can be further optimized to $O(\log(CK))$ by applying quantum adders and multipliers simultaneously. $\square$

**Quantum Batch Normalization Layer (qBatchNorm).**    The qBatchNorm layer, crucial for stabilizing training, is implemented in a hybrid quantum-classical manner. Due to the difficulty of computing global statistics (mean and variance) on a quantum state directly, we first perform a measurement on the state produced by the qConv layer. This DTM operation yields a classical snapshot of the feature map data. From this classical data, we compute the batch mean $\mu$ and variance $\sigma^2$. These classical parameters are then used to configure a quantum arithmetic circuit $U_{\mathrm{qBN}}$ within the QAM. This circuit applies the normalization transformation element-wise in superposition:

$$U_{\mathrm{qBN}} : |y\rangle \rightarrow |\gamma \frac{y - \mu}{\sqrt{\sigma^2 + \delta}} + \beta\rangle, \tag{14}$$

where $\gamma$ and $\beta$ are learnable classical parameters, and $|y\rangle$ is a register digitally encoding a single feature value. The complexity of the statistics estimation, which corresponds to the batch size $B'$, is $O(B'CK^2)$. The normalization, involving only single-qubit operations, costs $O(1)$.

**Quantum ReLU.**    The subsequent Quantum ReLU ($U_{\mathrm{QReLU}}$) is similarly implemented as an arithmetic comparison circuit within the QAM, applying $|y\rangle \rightarrow |\max(0, y)\rangle$.

Combining the discussion above naturally gives the overall complexity of qResNet.

*Proof of Theorem 3.2.* The convolution operation is the main bottleneck, which costs $\tilde{O}(CK^2)$ by Lemma B.1. Considering the sampling overhead, the overall gate complexity is then $\tilde{O}(CK^2 \times S(B, C, H, W))$. Furthermore, the input models are respectively queried for $X$ and $K$ twice in each implementation, since one is required for uncomputing. $\square$

---

**Algorithm 2:** Quantum Residual Block

---

**Require:** Input quantum state $|\psi_X\rangle$; Classical kernel $K$; Parameters $\gamma, \beta$.
**Ensure :** Output quantum state $|\psi_{\text{out}}\rangle$.

```
  /* Main Path                                                        */
1 |ψ_conv⟩ ← U_qConv(K)|ψ_X⟩;
  /* Hybrid Batch Normalization Step                                  */
2 X_conv ← Measure(|ψ_conv⟩);
3 μ, σ² ← ComputeBatchStats(X_conv);
4 |ψ_BN⟩ ← U_qBN(μ, σ², γ, β)|ψ_conv⟩;
5 |ψ_main⟩ ← U_QReLU|ψ_BN⟩;
  /* Shortcut and Addition                                            */
6 |ψ_shortcut⟩ ← |ψ_X⟩;
7 |ψ_sum⟩ ← QuantumAdd(|ψ_main⟩, |ψ_shortcut⟩);
  /* Final Activation                                                 */
8 |ψ_out⟩ ← U_QReLU|ψ_sum⟩;
9 return |ψ_out⟩;
```

---

### B.3 QUANTUM TRANSFORMER

This section details the quantum implementation of the Transformer architecture, which, like the quantum ResNet, is constructed from modular components. It leverages the Quantum Arithmetic Module (QAM) and the Quantum Linear Algebra Module (QLA). Without loss of generality, our analysis concentrates on a single batch and one-bit data. We assume that the input tensor of each building block $X^{(in)} \in \mathbb{R}^{N \times d}$ is quantum digitally encoded by the operator $O_X$, which is a quantum digital encoding of the input tensor $X^{(in)}$:

$$O_X |i\rangle |0^{\otimes d}\rangle = |i\rangle \bigotimes_j |X_{i,j}^{(in)}\rangle. \tag{15}$$

where $N$ is the sequence length and $d$ is the embedding dimension. For long-context tasks where $N \gg d$, the key to efficiency lies in how these modules handle the different dimensions: the large dimension $N$ is parallelized over using index registers, while the smaller dimension $d$ is processed arithmetically.

#### B.3.1 QUANTUM MULTI-HEAD SELF-ATTENTION MECHANISM

At the core of the quantum encoder, the self-attention mechanism is a hybrid of QAM-based arithmetic for local operations and QLA-based matrix multiplication for the final aggregation. The parallelism over the sequence length $N$ is achieved by encoding the token indices into dedicated quantum registers, allowing the QAM to operate on all elements in superposition.

**Q, K, V Projection and Score Calculation.** The initial step projects the input state $|X^{(in)}\rangle$ into Query ($Q$), Key ($K$), and Value ($V$) representations. This is $N$ independent multiplications on the $d$-dimensional vectors. This is performed by the QAM, conditioned on an index register $|i\rangle$ spanning the $N$ tokens. Considering the $h$ heads, the output quantum circuit $O_Q$ (similarly for $K, V$) should be

$$O_Q |i_N, i_h\rangle |0\rangle = |i_N, i_h\rangle \bigotimes_{0 \leq j < d_k} |Q_{i_h, i_N, j}\rangle, \tag{16}$$

where $i_N, i_h, j$ are respectively the sequence, head, and dimension indices. Given the query weight matrix encoded by $O_{W,q}$:

$$O_{W,q} |i_h\rangle |0\rangle = |i_h\rangle \bigotimes_{\substack{0 \le j < d_k, \\ 0 \le k \le d}} |(W_Q)_{j+i_h h, k}\rangle \equiv |i_h\rangle |W_{Q,i_h}\rangle, \qquad (17)$$

where $d_k = d/h$, $O_Q$ can be realized as follows:

$$|i_N, i_h\rangle |0\rangle |0\rangle |0^{\otimes d_k}\rangle$$

$$\xrightarrow{\text{Data Loading}} |i_N, i_h\rangle \bigotimes_{0 \le j < d} |X_{i_N,j}^{(in)}\rangle \bigotimes_{\substack{0 \le k < d_k, \\ 0 \le l \le d}} |(W_Q)_{kl}\rangle |0^{\otimes d_k}\rangle$$

$$\xrightarrow{\text{Element wise multiplication}} |i_N, i_h\rangle |X_{i_N}^{(in)}\rangle |W_{Q,i_h}\rangle \bigotimes_{\substack{0 \le \alpha < d_k, \\ 0 \le \beta \le d}} |(W_Q)_{\alpha+i_h h, \beta} X_{i_N,\beta}^{(in)}\rangle |0^{\otimes d_k}\rangle \qquad (18)$$

$$\xrightarrow{s \text{ steps addition}} |i_N, i_h\rangle |X_{i_N}^{(in)}\rangle |W_{Q,i_h}\rangle |W \circ X\rangle \bigotimes_{0 \le \alpha < d_k} |\sum_{0 \le \beta < d} (W_Q)_{\alpha+i_h h, \beta} X_{i_N,\beta}^{(in)}\rangle$$

$$\xrightarrow{\text{Rewriting \& Uncomputing}} |i_N, i_h\rangle \bigotimes_{0 \le \alpha < d_k} |Q_{i_N, \alpha+i_h h}\rangle \equiv |i_N, i_h\rangle |Q_{i_h, i_N}\rangle,$$

where $|W \circ X\rangle$ are simplified notations of the states generated in the second step. The derivation is almost the same for $K, V$.

The subsequent calculation of the attention scores, $\boldsymbol{S} = \boldsymbol{Q}\boldsymbol{K}^T/\sqrt{d_k}$, which results in an $N \times N$ matrix, follows a similar procedure. To compute all $N^2$ scores in parallel, we use two index registers, $|i\rangle$ and $|j\rangle$. An arithmetic circuit within the QAM then executes the dot product conditioned on these indices. The transformation on the quantum state can be abstractly represented as:

$$U_{\text{dot-prod}} : |i, j, i_h\rangle |Q_{i,i_h}\rangle |K_{j,i_h}\rangle |0\rangle \to |i, j\rangle |Q_{i,i_h}\rangle |K_{j,i_h}\rangle |S_{ij,i_h}\rangle. \qquad (19)$$

Here, the state $|i, j\rangle$ acts as a control, specifying which dot product to compute, while the operation itself happens on the data registers. The states $|Q_i\rangle$ and $|K_j\rangle$ represent the necessary data for the computation, which come from the previous discussion. This explicitly shows how the large $N \times N$ dimensional workload is handled through quantum parallelism rather than matrix size.

The cost of the preparation of $K, Q$ is simply twice the single cost, which is $O(d_k d) = O(d^2)$, equal to the total number of components involved. The circuit of the attention scores computation implements the inner product of dimension $d_k$, whose complexity is $O(d_k)$. They constitutes the overall $O(d^2)$ complexity.

**Softmax**   A full quantum implementation of the softmax function is notoriously difficult. We therefore adopt a hybrid quantum-classical approach. The state encoding the unnormalized score matrix $\boldsymbol{S}$ (as constructed in Eq. 19) is measured using the DTM. The $N \times N$ matrix of scores is then post-processed classically to compute the final attention matrix $\boldsymbol{A} = \text{softmax}(\boldsymbol{S})$. After uncomputing, we have built the arithmetic circuit $U_{A,arith}$:

$$U_{A,arith} |i, j, i_h\rangle |0\rangle = |i, j, i_h\rangle |A_{i,j,i_h}\rangle. \qquad (20)$$

**Weighted Sum**   The next step is the product $\boldsymbol{A}\boldsymbol{V}$. Here, $\boldsymbol{A}$ is a large, dense $N \times N$ matrix. Given our assumption that $N \gg d$, this large matrix multiplication is precisely the task for which the QLA is designed. The classical matrix $\boldsymbol{A}$ is used to construct its block-encoding unitary, $U_A$, where we have

$$\langle 0| U_A |0\rangle = \frac{1}{N} \sum_{i_h} |i_h\rangle \langle i_h| \otimes A_{i_h}. \qquad (21)$$

Here $A_{i_h} = \text{softmax}(Q_{i_h} K_{i_h}^T/\sqrt{d_k})$. This $U_A$ is built by the basic dense block-encoding Gilyén et al. (2019) based on the arithmetic circuit $U_{A,arith}$:

$$(H \otimes I \otimes I)(\text{SWAP} \otimes I) U_{A,arith}^{\dagger} U_{dac} U_{A,arith} (H \otimes I \otimes I), \qquad (22)$$

where $U_{dac}$ is the quantum circuit transforming bit strings into amplitude as introduced in Mitarai et al. (2019). The QLA then efficiently applies this unitary to the quantum state encoding the $\boldsymbol{V}$ matrix, which comes similarly from the arithmetic circuit and $U_{dac}$. The gate complexity comes mainly from several queries to $U_{A,arith}$, and therefore remains invariant.

**Multi-Head Parallelism.** After the weighted sum step, we obtain the quantum circuit realizing

$$U_{ws} |i_h, i\rangle |0\rangle = |i_h, i\rangle \frac{1}{N} \sum_j (A_{i_h} V_{i_h})_{ij}. \tag{23}$$

Note that when the input state is

$$|\vec{1}\rangle \frac{1}{\sqrt{d_k}} \sum_{i_h} |i_h\rangle, \tag{24}$$

the multi-head mechanism is naturally realized, with the output being

$$U_{ws} |\vec{1}\rangle |i\rangle |0\rangle = \frac{1}{N\sqrt{d_k}} |i\rangle \sum_{i_h, j} (AV)_{i_h, ij}. \tag{25}$$

The final linear projection $\boldsymbol{W}_O$ can be processed as an operator applying to the amplitude using similar method as above.

### B.3.2 QUANTUM FEED-FORWARD NETWORK (FFN)

Each Transformer block contains a position-wise Feed-Forward Network (FFN), applied independently to each of the $N$ token positions. This sub-layer is implemented entirely using the QAM. The mechanism is identical to that in the attention layer: the FFN's arithmetic circuits (two linear maps and a QReLU) are conditioned on an index register $|i\rangle$ that spans all $N$ token positions, thus processing all tokens in parallel. A complete quantum Transformer block is then formed by enclosing both the multi-head attention and FFN modules within residual connections and layer normalization, which are also implemented as QAM-based arithmetic operations conditioned on the token index.

*Proof of Theorem 3.3.* An encoder layer contains two residual connection layers, an MHSA layer, and an FFN layer. The MHSA layer is the resource bottleneck. For the MHSA layer, the arithmetic part scales as $\tilde{O}(d^2)$. The overall gate complexity is then $\tilde{O}(d^2 \times S(B, N, d))$ after considering the sampling overhead. It requires 6 queries to $X$, where $U_{A,arith}$ and its conjugate costs 2, and the vector encoding of $V$ costs 2. □

---

**Algorithm 3:** Quantum Transformer Block

---

**Require:** Input quantum state $|\psi_X\rangle$ encoding the sequence $X$; Classical parameters $\theta_{\text{Attn}}, \theta_{\text{FFN}}$
 for all layers.
**Ensure :** Output quantum state $|\psi_{\text{out}}\rangle$ after one Transformer block.

1 $|\psi_{\text{attn}}\rangle \leftarrow \text{QuantumMultiHeadAttention}(|\psi_X\rangle, \theta_{\text{Attn}})$;
 /* Multi-Head Attention Sub-layer                              */
 /* Hybrid approach:  QAM for projections/scores, QLA for AV
   product.                                                     */
2 $|\psi_{\text{add1}}\rangle \leftarrow \text{QuantumAdd}(|\psi_X\rangle, |\psi_{\text{attn}}\rangle)$;
 /* Residual connection:  Position-wise Add via QAM.            */
3 $|\psi_{\text{norm1}}\rangle \leftarrow U_{\text{LayerNorm}}(|\psi_{\text{add1}}\rangle)$;
4 $|\psi_{\text{ffn}}\rangle \leftarrow U_{\text{FFN}}(|\psi_{\text{norm1}}\rangle, \theta_{\text{FFN}})$;
 /* Feed-Forward Sub-layer                                      */
5 $|\psi_{\text{add2}}\rangle \leftarrow \text{QuantumAdd}(|\psi_{\text{norm1}}\rangle, |\psi_{\text{ffn}}\rangle)$;
 /* Residual connection:  Position-wise Add via QAM.            */
6 $|\psi_{\text{out}}\rangle \leftarrow U_{\text{LayerNorm}}(|\psi_{\text{add2}}\rangle)$;
7 **return** $|\psi_{out}\rangle$;

---

## C QUANTUM-ACCELERATED BACKPROPAGATION

Training deep neural networks relies on backpropagation, which systematically computes the gradient of the loss function with respect to the model's weights. We propose a quantum-accelerated

approach for this process, where the core matrix operations of the chain rule are mapped to our QAM and QLA modules. The overall process remains hybrid: gradients are typically stored and updated classically, but their computationally expensive calculation is offloaded to the quantum processor.

To illustrate the principle, we consider the backward pass through a single linear layer, defined by the forward pass $\boldsymbol{Y} = \boldsymbol{W}\boldsymbol{X}$. Here, $\boldsymbol{W}$ is a $d \times d$ weight matrix and $\boldsymbol{X}$ is a $d \times N$ matrix representing $N$ data points. Given the gradient from the subsequent layer, $\partial L / \partial \boldsymbol{Y}$ (a $d \times N$ matrix), we must compute two quantities: the gradient to be propagated backward, $\partial L / \partial \boldsymbol{X}$, and the gradient for updating the weights, $\partial L / \partial \boldsymbol{W}$.

**Gradient Calculation for Weights ($\partial L / \partial \boldsymbol{W}$): Handled by QLA.** The gradient with respect to the input is given by the chain rule:

$$\frac{\partial L}{\partial \boldsymbol{X}} = \boldsymbol{W}^T \frac{\partial L}{\partial \boldsymbol{Y}}. \tag{26}$$

This is a $(d \times d) \times (d \times N)$ matrix multiplication. Critically, this operation can be viewed as applying the small $(d \times d)$ matrix $\boldsymbol{W}^T$ to each of the $N$ columns of the incoming gradient $\partial L / \partial \boldsymbol{Y}$. This is a "position-wise" operation, perfectly suited for the QAM. By conditioning on an index register $|j\rangle$ spanning the $N$ columns, the QAM can perform all $N$ matrix-vector products in parallel, arithmetically processing the $d$-dimensional vectors in superposition.

**Gradient Calculation for Weights ($\partial L / \partial \boldsymbol{W}$): Handled by QLA.** The gradient with respect to the weights is an outer product:

$$\frac{\partial L}{\partial \boldsymbol{W}} = \frac{\partial L}{\partial \boldsymbol{Y}} \boldsymbol{X}^T. \tag{27}$$

This is a $(d \times N) \times (N \times d)$ matrix multiplication, resulting in a $d \times d$ gradient matrix. Each element $(\partial L / \partial \boldsymbol{W})_{ij}$ is the inner product of the $i$-th row of $\partial L / \partial \boldsymbol{Y}$ and the $j$-th row of $\boldsymbol{X}$. Both are vectors of length $N$. Given our assumption that $N \gg d$, these are high-dimensional inner products. This task is ideal for the QLA's inner product estimation capability Xiong et al. (2024). Instead of performing a full matrix multiplication, the QLA can be configured to efficiently estimate the $d^2$ required inner products between the corresponding $N$-dimensional quantum states, yielding the elements of the weight gradient.

This strategic division of labor is fundamental to our training approach. The QAM handles the backward flow of gradients through the network's data path by parallelizing over the sequence/batch dimension $N$. The QLA, in turn, handles the most intensive gradient calculations for weights, which involve contractions over this large $N$ dimension. This transforms the most demanding parts of backpropagation into potentially tractable quantum computations, paving the way for end-to-end quantum-accelerated training.

