# OpenReview forum: "Quantum Algorithm for Deep Neural Networks with Efficient I/O"
_ICLR.cc/2026/Conference — Submitted to ICLR 2026_

### Official Review · Reviewer_MrxZ · 2025-10-20

**Soundness:** 2
**Presentation:** 2
**Contribution:** 2
**Rating:** 4
**Confidence:** 3

**Summary:**

This paper proposes a hybrid quantum-classical framework for deep learning, which combines Quantum Linear Algebra (QLA) modules and Quantum Arithmetic Modules (QAM). The authors demonstrate how these modules can compose larger architectures such as Quantum ResNet and Quantum Transformer (qResNet / qTransformer). Besides, a Discrete Chebyshev Decomposition (DCD) protocol is introduced to mitigate inter-layer I/O bottlenecks. The paper claims that, under this modular design, large-scale matrix operations can be executed with asymptotic complexity $O(\mathrm{polylog}N)$, offering a potential quantum advantage in both forward and backward passes.

**Strengths:**

1. The split between QLA for handling high-dimensional linear algebra and QAM for handling low-dimensional arithmetic and nonlinearities is elegant and well motivated.

2. The DCD protocol is a technically meaningful idea. By approximating inter-layer quantum states with a truncated Chebyshev expansion, the DCD reduces measurement and reconstruction cost from $O(d^2)$ (state tomography) to $\tilde O(r/\delta)$, where $r \ll d$. This approach provides a solution to one of the most practical barriers in quantum deep learning, i.e. the I/O bottleneck, which is also numerically validated in Fig. 5.

3. The fidelity–accuracy study in Fig. 6 demonstrates that model accuracy remains stable even when quantum state fidelity drops from 0.99 to 0.9, which is an empirical validation of the good-enough principle.

**Weaknesses:**

1. The first problem is about the oracle construction complexity (without QRAM). The central assumption that the block-encoding unitary $U_A$ can be constructed and called in $O(\mathrm{polylog} N)$ time is not generally valid. This holds only if $A$ has a structured form (e.g., sparse or low-rank), or there exists a QRAM-like quantum oracle access to matrix entries. However, for dense and dynamical matrices, such as the attention matrices in Transformers, the paper does not provide a polylogarithmic-efficient block-encoding construction. The claimed polylogarithmic gate complexity implicitly assumes an oracle that is itself expensive to build from classical parameters. Without an explicit construction of $U_A$ independent of QRAM, the claimed quantum advantage remains hypothetical.

2. During training, the QLA operates on dynamical matrices such as attention matrices, which are parameterised. Therefore, its block-encoding directly depends on these parameters. So every update in the training stage implies a change in parameters and the the need to rebuild or recompile the corresponding oracle circuit. This reconstruction cost can be potentially expensive to negates any claimed quantum advantage once full training dynamics are considered, since the oracle must be regenerated each iteration.

**Questions:**

1. How is $U_A$ constructed from classical weights without assuming QRAM or sparsity? What is the actual gate complexity for a dense attention matrix?

2. When $W_Q, W_K, W_V$ in a quantum transformer are updated, how is $U_A(W)$ adjusted? Is there any incremental update mechanism, or must the oracle be recompiled entirely?

3. Given that gradients are computed and stored classically, do you consider the claimed quantum advantage to apply only to inference? If so, could this distinction be made explicit?

---

> ### Author Response · Authors · 2025-11-23
> **Response to Reviewer MrxZ (Part 1)**
>
> We thank the reviewer for the detailed and insightful review of our paper. We deeply appreciate your recognition of the strengths of our DCD protocol, the QLA/QAM modular design, and our fidelity analysis. Based on your constructive feedback, we have improved the manuscript as follows:
>
> ## Major changes
> * **Implementation details**. We have improved the computation details about tensor operations, quantum ResNet, and Transformer in Appendix A.2, B.2, and B.3. The step-by-step construction is detailed.
> * **Input model details**: We have provided the existing resource analysis of the general quantum input model in Appendix A.4.
> * **Training clarification**. We have clarified that the inference quantum advantage has been carefully examined in Section 4, whether quantum training can show a practical quantum advantage remains an open question.
>
> ## Weaknesses
>
> > The first problem is about the oracle construction complexity (without QRAM). The central assumption that the block-encoding unitary $U_A$ can be constructed and called in $O(\text{polylog}N)$ time is not generally valid. This holds only if it has a structured form (e.g., sparse or low-rank), or there exists a QRAM-like quantum oracle access to matrix entries. However, for dense and dynamical matrices, such as the attention matrices in Transformers, the paper does not provide a polylogarithmic-efficient block-encoding construction. The claimed polylogarithmic gate complexity implicitly assumes an oracle that is itself expensive to build from classical parameters. Without an explicit construction of $u_A$ independent of QRAM, the claimed quantum advantage remains hypothetical.
>
> Thank you for highlighting this critical issue regarding the complexity of $U_A$ construction. We respectfully clarify that our resource analysis explicitly assumes the availability of a Quantum Random Access Memory (QRAM) (specifically, the Quantum Input Model) to load classical data. We do not claim to bypass the need for QRAM; rather, we consider it a necessary prerequisite for achieving asymptotic speedup in processing large-scale classical data on quantum devices. In Appendices B.2 and B.3, we have explicitly adopted the Quantum Input Model in our revised manuscript.
>
> In Appendix A.4, we newly included a detailed resource analysis that accounts for the QRAM overhead. While the physical circuit size (hardware cost) of QRAM scales linearly as $O(N)$, the query depth (time complexity) scales as $O(\text{polylog}N)$ (e.g., using Select-Swap architectures, Clader, B. David, et al. "Quantum resources required to block-encode a matrix of classical data."). Therefore, the time-domain quantum advantage is preserved.
>
> A key contribution of our hybrid QLA/QAM framework is the significant reduction in the number of QRAM queries required (minimized to constant times per block), thereby mitigating the practical overhead of this component.
>
> > During training, the QLA operates on dynamical matrices such as attention matrices, which are parameterised. Therefore, its block-encoding directly depends on these parameters. So every update in the training stage implies a change in parameters and the the need to rebuild or recompile the corresponding oracle circuit. This reconstruction cost can be potentially expensive to negates any claimed quantum advantage once full training dynamics are considered, since the oracle must be regenerated each iteration.
>
> Thank you for this insightful comment. We agree that if the circuit topology required reconfiguration for every weight update, the recompilation cost would be prohibitive. However, our proposed implementation avoids this by treating parameters as inputs (data) rather than circuit structure (code).
> Specifically, we employ a scheme where the weight matrix $\mathbf{W}$ is digitally encoded into a quantum state (e.g., $|\mathbf{W}\rangle$). The core computational circuit (comprising quantum adders and multipliers in the QAM) remains structurally invariant. During training, an update to $\mathbf{W}$ only changes the classical bits fed into the state preparation routine. Since $\mathbf{W}$ typically involves lower dimensions ($d \times d$) compared to the input sequence $N$, efficiently preparing the state $|\mathbf{W}\rangle$ does not introduce a bottleneck, and no costly circuit topological recompilation is required.

---

> ### Author Response · Authors · 2025-11-23
> **Response to Reviewer MrxZ (Part 2)**
>
> ## Questions
>
> > How is $U_A$ constructed from classical weights without assuming QRAM or sparsity? What is the actual gate complexity for a dense attention matrix?
>
> Thank you for your insightful comment regarding the construction of $U_A$ for dense attention matrices and its associated gate complexity.
> We have updated Appendix B.3 to provide a detailed breakdown of the $U_A$ construction.
> To clarify: we do rely on a QRAM-like access model to achieve efficiency for dense matrices. We do not claim polylogarithmic construction from classical weights without this assumption.
> Regarding complexity:
> The complexity of the encoder block operations (projections via QAM and $U_A$ application via QLA) is $\tilde{O}(d^2)$, as stated in Theorem 3.3. The cost of QRAM queries to prepare the attention matrix state must be added. Based on standard QRAM architectures (e.g., Clader et al., 2022), for a dataset of size $N$, the circuit depth (time) is $O(\text{polylog}N)$, while the ancilla/gate count (space) is $O(N)$.
> Our analysis confirms that even with this overhead, the logarithmic time scaling with respect to $N$ is maintained.
>
> > When $W_Q,W_K,W_V$ in a quantum transformer are updated, how is $U_A(W)$ adjusted? Is there any incremental update mechanism, or must the oracle be recompiled entirely?
>
> Thank you for your concern for parameter updating and circuit recompilation. We respectfully clarify that the oracle does not require structural recompilation. As detailed in our response to Weakness 2, the weights $W_Q, W_K, W_V$ are handled via digital encoding. When parameters update, we simply update the classical values loaded into the quantum registers. The sequence of quantum gates (adders/multipliers) processing these registers remains fixed. Thus, the "update" is merely a change in the input state preparation, which is an efficient operation.
>
>
> > Given that gradients are computed and stored classically, do you consider the claimed quantum advantage to apply only to inference? If so, could this distinction be made explicit?
>
> You are correct that our primary numerical experiments and resource estimations in this paper validate the quantum advantage specifically for the inference phase. We have revised the Experiments sections to explicitly state this scope. However, the framework is not limited to inference. In Appendix C, we provide a theoretical derivation for Quantum-Accelerated Backpropagation, demonstrating how our QLA/QAM modules can be repurposed to compute gradients efficiently.
>
> We advocate for a "Classical Training, Quantum Inference" paradigm as a practical near-term path to advantage. As noted in recent literature (e.g., Huang et al., 2025), leveraging classical resources for optimization while using quantum processors for complex inference is a robust strategy to bypass trainability issues while harnessing quantum speedups.
>
> We believe these additions provide the necessary clarity for reproducing and verifying our proposed architecture.

---

> ### Comment · Reviewer_MrxZ · 2025-11-26
> **Question on the Author reply**
>
> If the polylogarithmic time complexity, measured by circuit depth, is considered to be “efficient” in the QRAM setting, then an analogous polylogarithmic time can also be achieved by classical data structures such as binary trees. Similar classical oracle models already appear in dequantized algorithms (e.g., in work by Ewin Tang [1]). For this reason, I am highly skeptical of the claimed quantum advantage.
>
> [1]. Ewin Tang. A quantum-inspired classical algorithm for recommendation systems. STOC 2019.

---

> ### Author Response · Authors · 2025-12-01
>
> We thank the reviewer for raising this critical question regarding the validity of quantum advantage in the presence of QRAM-based input models and quantum-inspired classical algorithms (dequantization). We respectfully clarify that quantum-inspired results do not invalidate our work, for three main reasons: theoretical complexity considerations, the practical overhead of dequantized algorithms compared with our specific proposal, and standard practices in the field.
>
> QRAM and quantum-inspired algorithms do not always negate quantum advantage. While we acknowledge the significance of Ewin Tang’s work [1–4] on dequantized algorithms for low-rank matrices, the existence of efficient classical data structures (such as binary trees) does not imply that all quantum speedups in this setting can be replicated classically. As argued in the foundational HHL paper [10], assuming that a classical algorithm can efficiently solve linear algebra problems (e.g., matrix inversion) simply because the input is accessible via a QRAM-like data structure would imply that $BQP \subseteq BPP$. This containment is widely believed to be unlikely. Moreover, "dequantized" algorithms specifically rely on low-rank structure assumptions, which do not cover the full range of cases where quantum algorithms (such as our QLA-based approach) provide speedups.
>
> Crucially, the polynomial-time guarantees of dequantized algorithms often hide substantial overheads in terms of precision $\epsilon$, rank $k$, and condition number $\kappa$. For instance, as analyzed in recent frameworks for dequantizing quantum machine learning (e.g., sampling-based sublinear low-rank matrix arithmetic frameworks), the complexity can scale prohibitively, such as $O(\epsilon^{-6} \|A\|_F^6)$ for singular value transformation, or a dependence of order $O(\sigma^{-28})$ for matrix inversion (where $\sigma$ denotes the smallest nonzero singular value).
>
> Even when taking quantum-inspired baselines into account, our work demonstrates distinct advantages. Unlike the high-degree polynomial dependence on error and rank found in dequantized algorithms, our Quantum Linear Algebra (QLA) modules exhibit significantly lower gate complexity. Our primary contribution is to address the scalability of deep neural networks (rather than focusing solely on a single linear algebra primitive). We introduce the Discrete Chebyshev Decomposition (DCD) protocol specifically to mitigate the I/O bottleneck—a challenge that dequantized algorithms do not resolve. Crucially, we go beyond purely asymptotic statements: our rigorous resource estimations (Section 4.3 and Table 2) quantitatively demonstrate that our hybrid model achieves practical efficiency gains over fully quantum approaches and remains competitive with classical baselines once the large hidden constants in dequantized algorithms are taken into account.
>
> Finally, the use of oracle/QRAM-based input models to explore quantum speedups remains a standard and necessary practice in theoretical quantum computing, particularly when analyzing algorithms whose running time is linear (or sublinear) in the input dimension. Recent works published in top-tier conferences continue to adopt similar input models to establish theoretical bounds for quantum algorithms in optimization and learning tasks. For example, Liu et al. [5] (NeurIPS 2025), Li et al. [6] (NeurIPS 2025), Luo et al. [7] (ICML 2025), Liu et al. [8] (ICML 2025), and Gong et al. [9] (ICLR 2025) all rely on oracle or QRAM-like access assumptions in their analyses.
>
>
> > [1] Tang, Ewin. "A quantum-inspired classical algorithm for recommendation systems." Proceedings of the 51st annual ACM SIGACT symposium on theory of computing. 2019.
>
> > [2] Arrazola, Juan Miguel, et al. "Quantum-inspired algorithms in practice." arXiv preprint arXiv:1905.10415 (2019).
>
> > [3] Tang, Ewin. "Quantum principal component analysis only achieves an exponential speedup because of its state preparation assumptions." Physical Review Letters 127.6 (2021): 060503.
>
> > [4] Chia, Nai-Hui, et al. "Sampling-based sublinear low-rank matrix arithmetic framework for dequantizing quantum machine learning." Journal of the ACM 69.5 (2022): 1-72.
>
> > [5] Chengchang Liu, "Quantum Speedups for Minimax Optimization and Beyond", NeurIPS 2025.
>
> > [6] Tongyang Li, "Near-Optimal Quantum Algorithms for Computing (Coarse) Correlated Equilibria of General-Sum Games", NeurIPS 2025.
>
> > [7] Bin Luo, "Quantum Algorithms for Finite-horizon Markov Decision Processes", ICML 2025.
>
> > [8] Chenghua Liu, "Quantum Speedup for Hypergraph Sparsification", ICML 2025.
>
> > [9] Weiyuan Gong, "Robustness of Quantum Algorithms for Nonconvex Optimization", ICLR 2025.
>
> > [10] Harrow, "Quantum algorithm for linear systems of equations", Physical review letters 2009 (103).

---

> ### Author Response · Authors · 2025-12-03
>
> We sincerely appreciate Reviewer MrxZ's recognition of our Discrete Chebyshev Decomposition (DCD) and thank them for the constructive feedback. We believe our clarifications on complexity claims and quantum advantage strengthen the value of this work.
>
> ### Summary of Responses
>
> 1. Oracle Complexity & QRAM
>    We provided a rigorous resource analysis under the standard *Quantum Input Model*, distinguishing between physical circuit size and time complexity. This confirms that the claimed speedup is preserved in the time domain.
>
> 2. Dynamic Weight Updates
>    We clarified that weights are treated as digital state inputs rather than fixed circuit gates, thus avoiding recompilation overhead during training.
>
> 3. Classical Dequantized Algorithms
>    We highlighted fundamental differences in scaling, showing that current dequantized algorithms incur larger overheads. Our approach aligns with community standards as seen in recent top-tier publications.
>
> ### Closing Remark
> Our work bridges theoretical quantum algorithms and practical deep learning. By quantifying resources under realistic QRAM assumptions and proposing a “Classical Training, Quantum Inference” paradigm, we present a tangible path toward near‑term quantum advantage while solving the critical I/O bottleneck.

---

### Official Review · Reviewer_JXSg · 2025-10-31

**Soundness:** 3
**Presentation:** 3
**Contribution:** 2
**Rating:** 6
**Confidence:** 3

**Summary:**

This paper proposes a promising path toward scalable quantum deep networks. The proposed Discrete Chebyshev Decomposition protocol could be a potential enabler of deep architectures on future fault-tolerant quantum hardware

The proposed framework is based on the assumption that similar performance can be achieved by a deep learning method when "rough" copies of the data are given.

A novel data transfer protocol is proposed that allow breaking down the task into subroutines.

The major finding is that the cost required to maintain a target accuracy scales sublinearly with the input dimension.

**Strengths:**

- the finding that the cost required to maintain a target accuracy scales sublinearly with the input dimension
- clear motivation and relevance
- a novel data transfer protocol
- theoretical foundation

**Weaknesses:**

- the good-enough assumption (approximate inter-layer information) is plausible but under-theorized
- only numerical simulations are reporte in the experiments

**Questions:**

Could you discuss the sensitivity of you r proposal to the choice of truncation rank r?

---

> ### Author Response · Authors · 2025-11-23
> **Response to Reviewer JXSg**
>
> We thank the reviewer for the detailed and insightful review of our paper. We deeply appreciate your recognition of the strengths of our DCD protocol, the QLA/QAM modular design, and our fidelity analysis. Based on your constructive feedback, we have improved the manuscript as follows:
>
> ## Weaknesses
> > the good-enough assumption (approximate inter-layer information) is plausible but under-theorized
>
> We appreciate the reviewer’s rigorous perspective. We acknowledge that while we provided empirical validation, the theoretical characterization of why deep networks are robust to this specific type of quantum approximation can be deepened. However, we respectfully clarify that the "good-enough" assumption is introduced as a guiding principle and intuition for our framework design. It posits that strict, high-fidelity state replication is often redundant, as deep models primarily rely on salient, low-frequency features to maintain performance. This principle is empirically grounded in the extensive success of classical model compression techniques (e.g., pruning and quantization). While deriving a complete learning-theoretic bound under this specific sampling noise remains a non-trivial open problem for the field, our work utilizes this principle to demonstrate that quantum protocols can be designed to exploit this inherent redundancy.
>
> > only numerical simulations are reported in the experiments
>
> We fully acknowledge that our experiments are simulation-based. This choice is necessitated by the specific scope of our research: our framework aims to solve the scalability issues of deep architectures (ResNet/Transformer) on future Fault-Tolerant Quantum Computers (FTQC), rather than optimizing for near-term (NISQ) devices. The depth and qubit requirements for a full Quantum Transformer significantly exceed current hardware capabilities. Furthermore, a primary contribution of this work is the asymptotic scaling advantage (e.g., sublinear measurement cost). Scaling laws cannot be reliably observed on small-scale, noisy hardware. Precise numerical resource analysis allows us to verify these benefits (Theorems 3.2 & 3.3) rigorously in a noise-free logical simulation environment, providing a clear blueprint for future hardware implementation.
>
>
> ## Questions
>
> > Could you discuss the sensitivity of you $r$ proposal to the choice of truncation rank $r$?
>
> Thank you for this important question regarding hyperparameter sensitivity. Our experiments demonstrate that the behaviors of the two models differ specifically. For the Transformer, we observe a "step-wise" improvement pattern: accuracy rises rapidly in several specific rank regions (not limited to the initial phase) while remaining significantly flatter in others, indicating that information is captured in discrete subspaces6. In contrast, ResNet exhibits a standard smooth convergence curve, where the accuracy improves consistently and the marginal gain diminishes as the rank increases. In both cases, this behavior suggests that $r$ acts as a flexible "knob" to trade off between accuracy and speedup without suffering from instability. We have included Table 2, which quantitatively lists the accuracy across different ranks, to further substantiate this stability.
>
> We believe these additions provide the necessary clarity for reproducing and verifying our proposed architecture.

---

> ### Author Response · Authors · 2025-12-03
>
> We sincerely appreciate Reviewer JXSg's recognition of our novel data transfer protocol and their view of this work as a promising path toward scalable quantum deep learning. We thank them for the constructive feedback on our theoretical assumptions and the reliance on simulations, which has helped us clarify the robustness and scope of our contributions.
>
> ### Summary of Responses
>
> 1. Simulation vs. Real Hardware
> We emphasized that our framework targets Fault‑Tolerant Quantum Computing (FTQC), where running deep architectures is infeasible on current NISQ devices. Numerical simulations are essential for verifying asymptotic scaling advantages in a noise‑free logical setting, offering a blueprint for future hardware.
>
> 2. “Good‑Enough” Approximation Assumption
> We contextualized this assumption within the established success of classical model compression techniques such as pruning and quantization. While full learning‑theoretic bounds for deep networks remain an open problem, our empirical evidence demonstrates that exploiting redundancy enables meaningful quantum speedups.
>
> 3. Sensitivity to Truncation Rank $r$
> We analyzed the effect of rank $r$ across architectures, showing discrete step‑wise gains for Transformers and smooth convergence for ResNets. Table 2 quantifies this flexibility, confirming that $r$ is a stable parameter to balance accuracy and efficiency.
>
> ### Closing Remark
> By grounding our design in proven compression principles, validating scalability through FTQC‑oriented simulations, and providing concrete analyses for both Transformer and ResNet models, this work delivers a robust protocol to alleviate the inter‑layer data transfer bottleneck in quantum deep learning. We believe these clarifications strengthen its theoretical foundation and practical relevance for the community.

---

### Official Review · Reviewer_9jeZ · 2025-10-31

**Soundness:** 4
**Presentation:** 3
**Contribution:** 4
**Rating:** 8
**Confidence:** 5

**Summary:**

This paper introduces a hybrid quantum-classical framework for designing deep quantum neural networks (QDNNs), addressing two of the field's most pressing challenges: the construction of deep architectures and the quantum data I/O bottleneck. The authors propose a novel data transfer mechanism, the Discrete Chebyshev Decomposition (DCD) protocol, shown to reduce overhead compared to standard tomography methods. The feasibility of these proposals is supported through concrete quantum implementations of ResNet and the Transformer. The paper is supported by a comprehensive suite of numerical experiments and resource analyses that validate its claims of favorable scaling and a tangible quantum advantage in specific, well-defined computational regimes.

**Strengths:**

The work is guided by the "good-enough" principle—arguing that perfect, high-fidelity state reconstruction between layers is unnecessary—provides a powerful justification for mitigating the I/O bottleneck. The modular architecture is an innovative solution to managing quantum overhead.

The proposed DCD protocol addresses the critical data transfer bottleneck. The numerical evidence for its sublinear scaling of measurement cost with input dimension is a significant result.

The paper support the conclusions thorough sufficient empirical validation. The authors provide not only a comparison of DCD against standard tomography but also a detailed resource analysis comparing their hybrid model to a fully QLA-based baseline.

**Weaknesses:**

The entire framework's efficacy, and particularly the success of the DCD protocol, is predicated on the core assumption that intermediate quantum states are highly compressible. While the numerical results strongly support this hypothesis for the tested models, its universality is an open question. It would significantly strengthen the paper if the authors discussed the potential failure modes of this assumption.

The proposed algorithms are designed for a fault-tolerant quantum computer, placing them in a long-term research context. While a detailed resource compilation is beyond the scope of this work, the paper would benefit from a brief, order-of-magnitude discussion on the required hardware scale.

While the high-level architectural descriptions  are clear, the paper could be improved by providing more fine-grained implementation details for some of the key quantum modules.

**Questions:**

The entire framework's efficacy, and particularly the success of the DCD protocol, is predicated on the core assumption that intermediate quantum states are highly compressible. While the numerical results strongly support this hypothesis for the tested models, its universality is an open question. It would significantly strengthen the paper if the authors discussed the potential failure modes of this assumption.

The proposed algorithms are designed for a fault-tolerant quantum computer, placing them in a long-term research context. While a detailed resource compilation is beyond the scope of this work, the paper would benefit from a brief, order-of-magnitude discussion on the required hardware scale.

While the high-level architectural descriptions  are clear, the paper could be improved by providing more fine-grained implementation details for some of the key quantum modules.

---

> ### Author Response · Authors · 2025-11-23
> **Response to Reviewer 9jeZ**
>
> We thank the reviewer for the detailed and insightful review of our paper. We deeply appreciate your recognition of the strengths of our DCD protocol, the QLA/QAM modular design, and our fidelity analysis. Based on your constructive feedback, we have improved the manuscript as follows:
>
> ## Weaknesses & Questions
>
> > The entire framework's efficacy, and particularly the success of the DCD protocol, is predicated on the core assumption that intermediate quantum states are highly compressible. While the numerical results strongly support this hypothesis for the tested models, its universality is an open question. It would significantly strengthen the paper if the authors discussed the potential failure modes of this assumption.
>
> Thank you for this valuable suggestion. We agree that discussing the boundaries of our assumption strengthens the paper. Our hypothesis aligns with the concept of "Spectral Bias" in classical deep learning theory, which posits that neural networks prioritize learning low-frequency, globally salient functions. This explains why the "good-enough" principle holds for standard tasks like image classification.
>
> We have added a discussion in the revised manuscript (Section 5) acknowledging that this assumption may face challenges in domains dominated by high-frequency, high-entropy features, such as chaotic physical system simulation, high-frequency financial time series analysis, or cryptographic data processing. In these scenarios, the salient information is not concentrated in the low-rank subspace, necessitating a much higher rank $r$ in the DCD protocol, which would reduce the computational speedup. We identify the rigorous characterization of these boundaries as critical future work.
>
> > The proposed algorithms are designed for a fault-tolerant quantum computer, placing them in a long-term research context. While a detailed resource compilation is beyond the scope of this work, the paper would benefit from a brief, order-of-magnitude discussion on the required hardware scale.
>
> We appreciate this suggestion. In Section 4, we provided estimates regarding circuit depth and sampling overhead. To address the hardware scale more comprehensively, we have added Appendix A.4. Based on the resource analysis of QRAM (referencing Clader et al., 2022), we estimate that the logical qubit requirement scales linearly with the input size, i.e., $\mathcal{O}(N)$. For a typical input dimension in our experiments (e.g., $N \approx 6.5 \times 10^4$ for a $256 \times 256$ image), this implies a requirement of approximately $10^5$ logical qubits for data storage. This confirms that our framework is indeed positioned for the mature Fault-Tolerant Quantum Computing (FTQC) era, where such resource scales are anticipated.
>
> > While the high-level architectural descriptions are clear, the paper could be improved by providing more fine-grained implementation details for some of the key quantum modules.
>
> Thank you for pointing this out. We have significantly expanded Appendix B.2 (Quantum ResNet) and Appendix B.3 (Quantum Transformer) in the revised manuscript. Specifically, we have added: Detailed logical flows for how tensor operations are mapped to quantum gates; Clarified how the QLA and QAM modules interface within a single layer; More specific descriptions of the block-encoding unitaries involved in the attention mechanism.
>
> We believe these additions provide the necessary clarity for reproducing and verifying our proposed architecture.

---

> > ### Comment · Reviewer_9jeZ · 2025-11-26
> >
> > I thank the authors for their detailed response and the comprehensive revisions.I am satisfied with how the authors addressed the validity of the compressibility assumption. By linking the DCD protocol to the "Spectral Bias" phenomenon in classical deep learning, the authors provide a convincing theoretical justification for why the low-rank approximation holds for standard tasks. I also appreciate the intellectual honesty in acknowledging potential limitations in high-entropy domains like chaotic systems or cryptography in the new Section 5.Regarding the hardware scale, the added Appendix A.4 provides necessary clarity. The estimation of $\mathcal{O}(N)$ scaling (approx. $10^5$ logical qubits) based on QRAM overhead confirms that the proposed framework is well-positioned for the FTQC era.With the additional implementation details in Appendix B covering the tensor mapping and attention mechanisms, my concerns have been fully resolved. I recommend acceptance.

---

> ### Author Response · Authors · 2025-12-03
>
> We sincerely appreciate Reviewer 9jeZ's strong endorsement of our Discrete Chebyshev Decomposition (DCD) protocol and their recognition of its value in solving the critical I/O bottleneck in quantum deep learning. We thank them for the constructive suggestions to refine the theoretical boundaries of our assumptions and clarify targeted hardware requirements, which have led to significant improvements in the manuscript.
>
> ### Summary of Responses
>
> 1. Universality of the “Good‑Enough” Assumption
>    We linked our compressibility hypothesis to the classical concept of *Spectral Bias*, explicitly identifying failure modes in high‑entropy domains such as chaotic simulation or cryptography. This delineates the conditions under which the framework applies and clarifies its boundaries.
>
> 2. Hardware Scale & Resource Estimation
>    We added Appendix A.4 with QRAM‑based resource analysis showing that logical qubit requirements scale linearly as $\mathcal{O}(N)$. For a $256 \times 256$ image input, this corresponds to approximately $10^5$ logical qubits, framing the proposal squarely within the mature FTQC era.
>
> 3. Expanded Implementation Details
>    We enhanced Appendix B.2 (ResNet) and B.3 (Transformer) with detailed module flows, QLA/QAM interface clarifications, and explicit block‑encoding unitary descriptions. This ensures reproducibility and concreteness for community adoption.
>
> ### Closing Remark
> By defining the practical limits of our “Good‑Enough” principle, quantifying hardware scale for FTQC readiness, and detailing deep architecture implementations, the revised manuscript offers a clear, reproducible blueprint for scalable quantum deep learning. We believe these clarifications strengthen both the theoretical rigor and practical utility of the work for future developments in the field.

---

### Official Review · Reviewer_yrP9 · 2025-11-06

**Soundness:** 2
**Presentation:** 2
**Contribution:** 2
**Rating:** 2
**Confidence:** 3

**Summary:**

This paper proposes a hybrid quantum-classical framework for constructing deep quantum neural networks (QDNNs), such as Quantum ResNets and Quantum Transformers. The core idea is to separate computations into Quantum Linear Algebra (QLA) modules for large-dimension ($N$) operations and Quantum Arithmetic Modules (QAMs) for small-dimension ($d$) operations. To address the I/O bottleneck between layers, the paper introduces the "good-enough" principle, postulating that intermediate states can be imperfectly reconstructed. This is implemented via a novel "Discrete Chebyshev Decomposition" (DCD) protocol, a lossy compression scheme designed to be more efficient than full tomography.

**Strengths:**

This work presents a framework to overcome two major obstacles in quantum deep neural networks: (1) constructing deep architectures (e.g., multi-layer ResNets and Transformers) in the quantum domain, and (2) the prohibitive quantum data I/O or state-preparation overhead. The authors introduce the “good-enough” principle: that a deep learning model can still perform well even when given “rough” copies of the input data. They propose a modular architecture that splits computations into quantum linear algebra modules (QLA) and quantum arithmetic modules (QAM), enabled by a novel data-transfer protocol called Discrete Chebyshev Decomposition (DCD). Their numerical validation suggests that the measurement cost to maintain target accuracy scales sublinearly with input dimension (thus preserving quantum advantage by mitigating I/O bottlenecks). They also provide a resource-analysis arguing that, on a fault-tolerant quantum computer, such quantum neural networks could be more scalable than classical counterparts.

**Weaknesses:**

The "Good-enough" Principle is an Unproven Hypothesis: The paper's central claim is that a deep network can "achieve similar performance when 'rough' copies of the data are allowed" (Abstract). This is implemented by the DCD protocol, which truncates the Chebyshev spectrum, effectively keeping only $r$ low-frequency components ($r \ll d$).

Lack of Justification: This principle is "believed" (Abstract) or "posited" (Introduction) rather than proven. There is no theoretical guarantee that the high-frequency information discarded by DCD is irrelevant. For many tasks, this high-frequency data (e.g., fine textures, specific tokens) is precisely what distinguishes different classes.

Weak Analogy: The comparison to classical pruning/quantization is misleading. Those methods are applied with care, often require retraining to compensate for information loss, and do not typically involve discarding the entire high-frequency spectrum of a layer's output.

Flawed Complexity Analysis and Hidden Scaling: The paper claims its QLA-QAM hybrid (Fig 1d) avoids the multiplicative complexity of fully QLA-based networks (Fig 1c). However, the proposed solution (DCD) requires a measurement-and-re-preparation cycle at every layer.

Re-introduction of Multiplicative Cost: This very cycle is a known bottleneck that decoheres the state and introduces a multiplicative sampling cost that scales with depth. The overall cost to execute $L$ layers is not additive.

Unrealistic Assumption of Fault-Tolerance: The paper explicitly targets fault-tolerant quantum computers (FTQC) (Abstract, Sec 1).

This assumption sidesteps all of the real challenges in quantum computing (noise, coherence, qubit count). The proposals are purely theoretical and have no path to near-term implementation.

The QAMs, in particular, which rely on complex quantum arithmetic (adders, multipliers) to implement non-linearities (like QReLU), would require an enormous number of gates and an exceptionally high degree of fault tolerance, making them practically infeasible even on projected early FTQCs. The paper glosses over this immense resource cost.

**Questions:**

While the paper is ambitious and addresses a meaningful barrier in quantum deep learning (the I/O bottleneck), several concerns remain:

1. Empirical evidence mostly simulation-based and limited scope

The results are reported from simulations rather than real quantum hardware; thus actual overheads (error rates, decoherence, state-preparation overhead, readout latency) are not demonstrated.

The claim of sublinear measurement scaling is intriguing but might depend heavily on idealised conditions (noise-free, fault-tolerant assumptions). It is unclear how robust this behaviour is under realistic quantum hardware constraints.

It is not fully clear how “input dimension” is defined in the experiments (e.g., image size, sequence length, parameter count) or whether the datasets used reflect real large-scale deep learning problems (e.g., full-sized Transformers, very large inputs).

2. “Good-enough” principle needs stronger justification and clarity

The idea that approximate or “rough” data copies suffice is appealing but not deeply analysed: under what error bounds or fidelity reductions does performance degrade? What kind of “roughness” is acceptable?

Without a more quantitative sensitivity or ablation study on the fidelity/approximation error vs accuracy trade-off, the principle remains somewhat heuristic.

3. Architectural and algorithmic details are underdeveloped

The mapping of classical deep network layers (ResNet, Transformer) into quantum modules is described at a high level but lacks full transparency: How exactly are QLA and QAM modules defined for, say, attention in Transformers? What is the exact workflow or quantum circuit structure?

State preparation and encoding (which is often the main bottleneck) are only addressed by the DCD protocol; it would help to compare DCD explicitly against existing state-preparation or classical-quantum hybrid I/O protocols.

The resource analysis is valuable, but details such as qubit counts, gate depths, measurement sampling cost, ancillary qubits, and error mitigation strategies are not fully fleshed out.

4. Comparison against classical deep learning and classical I/O bottleneck solutions is weak

Since the paper claims a quantum advantage over classical counterparts, there should be clearer benchmarking or discussion of comparable classical I/O mitigation strategies (e.g., sketching, approximation, sparsification) and how the quantum approach compares or improves on them.

It is not shown how the quantum architecture performs (or would perform) in practice compared to highly optimised classical deep learning pipelines under I/O constraints. Without that, the practical significance is harder to assess.

5. Scalability to near-term hardware is unclear

The paper posits a fault-tolerant quantum computer scenario, but near-term quantum devices (NISQ) have major limitations: gate fidelity, coherence times, qubit connectivity, readout noise. The gap between the idealised framework and actual hardware readiness is not sufficiently addressed.

The work would benefit from a discussion of how one might implement DCD and the modular architecture on upcoming quantum hardware (or hybrid classical-quantum systems) and what the minimum requirements are.

**Relevant References for Inclusion**

Here are key references the authors should cite to situate their work properly and acknowledge related work in quantum deep learning, quantum neural networks, and quantum I/O/data-preparation concerns:

1. Quantum deep learning / quantum neural network foundational works

- Beer, K. et al. (2020). \textit{Training deep quantum neural networks.} Nature Communications, 11, 808.
- Zhao, R. \& Wang, Z. (2021). \textit{A review of quantum neural networks: methods, models, and applications.} arXiv:2109.01840.
- Valdez, F. \& Melin, P. (2022). \textit{A review on quantum computing and deep learning algorithms and their applications.} Applied Intelligence (via PMC).

2. Quantum I/O, data encoding and quantum machine learning scalability

- Peral-García, D. et al. (2024). \textit{Systematic literature review: Quantum machine learning.} Journal of Computer Languages, 79, 102–?.
- Kwak, Y. (2023). \textit{Quantum distributed deep learning architectures: Models and possibilities.} Journal of Data and Information Science (special issue).
- Stein, S. A. et al. (2021). \textit{QuClassi: A Hybrid Deep Neural Network Architecture based on Quantum State Fidelity.} arXiv:2103.11307.

3. Deep network/architecture adaptation to quantum settings

- Levine, Y., Sharir, O., Cohen, N. \& Shashua, A. (2018). \textit{Quantum Entanglement in Deep Learning Architectures.} arXiv:1803.09780.
- Pan, X. et al. (2022). \textit{Deep quantum neural networks equipped with backpropagation on a superconducting processor.} arXiv:2212.02521.

4. Data preparation / state-preparation and quantum linear algebra subroutines

- Childs, A. M., Kothari, R. \& Somma, R. (2017). \textit{Quantum algorithm for systems of linear equations with exponentially improved dependence on precision.}
- Kerenidis, I. \& Prakash, A. (2016). \textit{Quantum recommendation systems.}

---

> ### Author Response · Authors · 2025-11-21
> **Response to Reviewer yrP9 (Part 1)**
>
> We thank the reviewer for the detailed and constructive feedback. We appreciate the recognition of our framework's potential to address the critical I/O bottleneck and construct deep architectures in Quantum Deep Learning. Based on your valuable suggestions, we have made the following improvements to the manuscript:
>
> ## Major changes
> * **Enhanced related works.** We have expanded Section 1.1 to include the recommended references, introducing contributions in this field and relations to our proposals.
> * **Clarified definitions.** We have explicitly explained the definition of the "input dimension $N$" in the Experiment section to avoid confusion ($N$ represents sequence length for Transformers and spatial pixel count $H \times W$ for ResNets).
> * **Implementation details.** We have improved the computation details about tensor operations, quantum ResNet, and Transformer in Appendix A.2, B.2, and B.3. The step-by-step construction is detailed.
>
> ## Weaknesses
> > The "Good-enough" Principle is an Unproven Hypothesis: The paper's central claim is that a deep network can "achieve similar performance when 'rough' copies of the data are allowed" (Abstract). This is implemented by the DCD protocol, which truncates the Chebyshev spectrum, effectively keeping only $r$ low-frequency components ($r\ll d$).
> Lack of Justification: This principle is "believed" (Abstract) or "posited" (Introduction) rather than proven. There is no theoretical guarantee that the high-frequency information discarded by DCD is irrelevant. For many tasks, this high-frequency data (e.g., fine textures, specific tokens) is precisely what distinguishes different classes.
> Weak Analogy: The comparison to classical pruning/quantization is misleading. Those methods are applied with care, often require retraining to compensate for information loss, and do not typically involve discarding the entire high-frequency spectrum of a layer's output.
>
> We thank the reviewer for raising this fundamental question regarding the validity of our core principle. We respectfully clarify that establishing the validity of the "Good-enough" principle is not merely an assumption, but one of the primary scientific contributions of this work. We designed the DCD protocol specifically to investigate the extent to which deep quantum networks can tolerate "rough" state approximations. Our experimental results (e.g., Figure 5) provide strong empirical evidence for this hypothesis: the fact that our DCD-based models recover high accuracy even with significantly truncated spectra confirms that the "Good-enough" principle holds for these deep architectures. The subsequent resource analysis, grounded in this successful approximation, further quantifies the practical quantum advantage.
>
> Regarding the concern about discarding high-frequency information: In modern deep learning theory, it is widely understood that while initial layers process high-frequency details (textures), deep semantic representations are inherently low-rank. Our analog is used to support the opinion that there is a lot of redundancy in deep neural networks, which has been observed and exploited in many ways, such as classical pruning, Low-Rank Adaptation (LoRA), or Dropout techniques. Therefore, the "high-frequency" components discarded by DCD often correspond to noise or redundant features rather than critical semantic signals, as evidenced by the "elbow effect" in our rank analysis (Figure 5e).

---

> ### Author Response · Authors · 2025-11-21
> **Response to Reviewer yrP9 (Part 2)**
>
> ## Weakness (continued)
>
> > Flawed Complexity Analysis and Hidden Scaling: The paper claims its QLA-QAM hybrid (Fig 1d) avoids the multiplicative complexity of fully QLA-based networks (Fig 1c). However, the proposed solution (DCD) requires a measurement-and-re-preparation cycle at every layer.
> Re-introduction of Multiplicative Cost: This very cycle is a known bottleneck that decoheres the state and introduces a multiplicative sampling cost that scales with depth. The overall cost to execute $L$ layers is not additive.
>
> We thank the reviewer for this crucial comment. We realize there is a misunderstanding regarding the specific definition of "multiplicative complexity" in the context of Quantum Linear Algebra (QLA) versus our hybrid approach. We clarify this as follows:
>
> In a fully coherent QLA network (Figure 1c), the nesting of block-encoding operations leads to a global normalization factor $\alpha_{total} = \prod \alpha_i$. This results in an exponential scaling of the resources required to amplify the amplitude for the final measurement. By introducing the measurement-and-re-preparation cycle, we intentionally break this coherent chain. Consequently, the complexity scales additively with the network depth $L$: the total runtime is the sum of the time required to sample and reconstruct each layer sequentially ($T_{total} = \sum T_{layer}$), successfully avoiding the exponential explosion of overheads associated with deep coherent circuits. We emphasize that our framework utilizes Quantum Arithmetic Modules (QAM) for feature-dimension operations. Unlike probabilistic QLA subroutines, QAMs are deterministic circuits composed of adders and multipliers. Their circuit depth scales additively with the number of operations. This structural choice, depicted in Figure 1d, ensures that intra-layer operations remain efficient and do not introduce additional probabilistic bottlenecks.
>
> We respectfully clarify that there is no "hidden scaling". Minimizing the measurement-and-re-preparation cost is a primary goal of this paper. Our complexity theorems explicitly include the sampling overhead term, such as $S(B,C,H,W)$ in Theorem 3.2. As shown in Figure 5 and Table 2, our total resource estimation ($Q$) incorporates both the circuit complexity AND the cumulative sampling costs. The results demonstrate that, even with these sampling overheads included, our method achieves significantly lower total resource consumption compared to the baseline, precisely because the DCD protocol drastically reduces the sampling term $S$ compared to full tomography.
>
> > Unrealistic Assumption of Fault-Tolerance: The paper explicitly targets fault-tolerant quantum computers (FTQC) (Abstract, Sec 1).
> This assumption sidesteps all of the real challenges in quantum computing (noise, coherence, qubit count). The proposals are purely theoretical and have no path to near-term implementation.
> The QAMs, in particular, which rely on complex quantum arithmetic (adders, multipliers) to implement non-linearities (like QReLU), would require an enormous number of gates and an exceptionally high degree of fault tolerance, making them practically infeasible even on projected early FTQCs. The paper glosses over this immense resource cost.
>
> We thank the reviewer for raising this critical issue. We agree that our proposal is purely theoretical and does not target current Near-Term Intermediate-Scale Quantum (NISQ) devices. We clarify the necessity of the FTQC assumption as follows:
>
> The primary goal of this work is to explore the asymptotic scalability of quantum deep neural networks, specifically focusing on achieving acceleration for the large input dimension $N$ commonly encountered in modern deep learning. Assuming FTQC is the only viable approach to assess the potential required for quantum algorithms to solve large-scale deep learning bottlenecks. NISQ algorithms such as Variational Quantum Circuits (VQCs) face the Barren Plateau problem, which limits the training and expressivity of deep networks, while their simulation is expoential hard, preventing reliable performance evaluation for large-scale deep learning tasks.

---

> ### Author Response · Authors · 2025-11-21
> **Response to Reviewer yrP9 (Part 3)**
>
> ## Questions
> > Empirical evidence mostly simulation-based and limited scope
> The results are reported from simulations rather than real quantum hardware; thus actual overheads (error rates, decoherence, state-preparation overhead, readout latency) are not demonstrated.
> The claim of sublinear measurement scaling is intriguing but might depend heavily on idealised conditions (noise-free, fault-tolerant assumptions). It is unclear how robust this behaviour is under realistic quantum hardware constraints.
>
> We thank the reviewer for this comment. Simulating full-scale deep Quantum Neural Networks (e.g., ResNet-18) on real quantum hardware is currently impossible due to qubit limitations. Our simulations serve as exact logical verifications of the algorithm's correctness and resource scaling laws. This is the standard validation method for FTQC algorithms (similar to how HHL or Shor's algorithm are analyzed before hardware is ready).
>
> > It is not fully clear how "input dimension" is defined in the experiments (e.g., image size, sequence length, parameter count) or whether the datasets used reflect real large-scale deep learning problems (e.g., full-sized Transformers, very large inputs).
>
> We thank the reviewer for noting this ambiguity. The conception of input dimension is further clarified while the datasets we used are CUB-200-2011, which is clarified at the beginning of Section 4.
>
> > "Good-enough“ principle needs stronger justification and clarity
> The idea that approximate or “rough” data copies suffice is appealing but not deeply analysed: under what error bounds or fidelity reductions does performance degrade? What kind of “roughness” is acceptable?
> Without a more quantitative sensitivity or ablation study on the fidelity/approximation error vs accuracy trade-off, the principle remains somewhat heuristic.
>
> We thank the reviewer for suggesting a sensitivity analysis. We actually performed this analysis in Figure 6. Figure 6 plots Classification Accuracy vs. Infidelity (state preparation error). It explicitly shows that the DCD method maintains high accuracy even when the state infidelity is relatively high (high "roughness"), whereas the standard tomography method degrades rapidly. This is the quantitative sensitivity study demonstrating the robustness of the principle.
>
> > Architectural and algorithmic details are underdeveloped
> The mapping of classical deep network layers (ResNet, Transformer) into quantum modules is described at a high level but lacks full transparency: How exactly are QLA and QAM modules defined for, say, attention in Transformers? What is the exact workflow or quantum circuit structure?
>
> We thank the reviewer for asking for more transparency. We have improved the computation details about tensor operations, quantum ResNet, and Transformer in Appendix A.2, B.2, and B.3. The step-by-step construction is detailed.
>
> >State preparation and encoding (which is often the main bottleneck) are only addressed by the DCD protocol; it would help to compare DCD explicitly against existing state-preparation or classical-quantum hybrid I/O protocols.
>
> We thank the reviewer for the comparative experiments. We have made such a comparison between our DCD protocol and a typical measurement and state-preparation scheme as depicted in Figures 5 and 6.
>
> > The resource analysis is valuable, but details such as qubit counts, gate depths, measurement sampling cost, ancillary qubits, and error mitigation strategies are not fully fleshed out.
>
> We thank the reviewer for requesting these critical quantitative details. We recognize the necessity of fully fleshed-out resource tables for FTQC algorithms. Our current resource analysis focuses on the asymptotic scaling of the total cost $Q$ for the entire network, which already explicitly incorporates the dominant measurement sampling cost $S$ (as detailed in Theorem 3.2 and Figure 5). We commit to calculating and integrating specific qubit counts and ancillary qubit requirements for the QLA and QAM subroutines into an expanded Appendix of the revised manuscript. We can now clarify that the total number of qubits required by this framework differs from the QRAM storage by at most a factor of a few.

---

> ### Author Response · Authors · 2025-11-21
> **Response to Reviewer yrP9 (Part 4)**
>
> > Comparison against classical deep learning and classical I/O bottleneck solutions is weak
> Since the paper claims a quantum advantage over classical counterparts, there should be clearer benchmarking or discussion of comparable classical I/O mitigation strategies (e.g., sketching, approximation, sparsification) and how the quantum approach compares or improves on them.
> It is not shown how the quantum architecture performs (or would perform) in practice compared to highly optimised classical deep learning pipelines under I/O constraints. Without that, the practical significance is harder to assess.
>
> ## Questions (continued)
>
> > Scalability to near-term hardware is unclear
> The paper posits a fault-tolerant quantum computer scenario, but near-term quantum devices (NISQ) have major limitations: gate fidelity, coherence times, qubit connectivity, and readout noise. The gap between the idealised framework and actual hardware readiness is not sufficiently addressed.
> The work would benefit from a discussion of how one might implement DCD and the modular architecture on upcoming quantum hardware (or hybrid classical-quantum systems) and what the minimum requirements are.
>
> We thank the reviewer for raising the NISQ implementation question. In this paper, we mainly focus on the algorithmic bottleneck in the FTQC era, and this framework is not designed for NISQ devices. Our contribution is for the theoretical scalability in the future research of quantum deep learning, distinct from near-term heuristic approaches like variational quantum algorithms (VQAs).
>
> ## Relevant References for Inclusion
>
> We thank the reviewer for the curated list of references. We have incorporated them into Section 1.1 and the Introduction to better situate our work within the broader landscape of Quantum Deep Learning foundations and I/O challenges.

---

> > ### Author Response · Authors · 2025-11-27
> > **Comment for Reviewer yrP9**
> >
> > Dear Reviewer yrP9:
> > Thank you for the valuable review and your time. In response to your feedback, we have diligently revised the paper to address the concerns raised. If any points remain unclear or if further adjustments are needed, we would be happy to receive your guidance and will address it promptly.
> >
> > Sincerely,
> >
> > The authors

---

> ### Author Response · Authors · 2025-12-03
>
> We sincerely thank Reviewer yrP9 for the constructive feedback and recognition of our theoretical contributions.
> We believe our clarifications on the Good-Enough Principle, complexity analysis, and FTQC assumption further strengthen the value of this work and its relevance to the community.
>
> ### Summary of Responses
>
> 1. Good-Enough Principle
>    We clarified that this is not an assumption but an empirically validated finding (Fig. 5, 6), consistent with established deep learning theory on the low‑rank nature and redundancy of deep representations.
>    Experiments confirm that QDNNs remain robust under the spectral truncation performed by our Discrete Chebyshev Decomposition (DCD) protocol.
>
> 2. Complexity & Measurement
>    We showed that the measurement–re‑preparation cycle does not incur multiplicative costs.
>    Instead, it breaks the coherent chain, converting exponential overhead into linear additive complexity. All sampling costs are explicitly accounted for in Theorem 3.2, demonstrating clear asymptotic advantage.
>
> 3. FTQC Assumption
>    Given the hardness of classically simulating quantum systems, we argue that a rigorous theoretical analysis based on FTQC is currently the only viable path to determining asymptotic scalability for large‑scale Quantum Deep Learning.
>
> 4. Additional Improvements
>    We added precise definitions for input dimensions and provided step‑by‑step quantum circuit constructions for Quantum ResNets and Quantum Transformers in the Appendix.
>    We have also included all requested literature comparisons for completeness
>
> ### Closing Remark
>
> While acknowledging the gap between current hardware and theory, we emphasize that the community urgently needs a scalable architectural blueprint that resolves the I/O bottleneck.
> Our work offers this roadmap, moving from heuristic NISQ attempts toward designs with rigorous asymptotic advantages, thus laying a foundation for practical, large‑scale Quantum Deep Learning.

---

### Author Response · Authors · 2025-11-21
**Revised manuscript uploaded (Dec. 3)**

## Revised manuscript (version 20251203)
We have submitted the revised manuscript with the following changes (see Supplementary Material for the highlighted version).

### Major changes
* **Enhanced related works.** We have expanded Section 1.1 to include the recommended references, introducing contributions in this field and relations to our proposals.
* **Clarified definitions.** We have explicitly explained the definition of the "input dimension $N$" in the Experiment section to avoid confusion ($N$ represents sequence length for Transformers and spatial pixel count $H \times W$ for ResNets).
* **Implementation details.** We have improved the computation details about tensor operations, quantum ResNet, and Transformer in Appendix A.2, B.2, and B.3. The step-by-step construction is detailed.
* **Input model details**: We have provided the existing resource analysis of the general quantum input model in Appendix A.4.
* **Training clarification**: We have clarified that the inference quantum advantage has been carefully examined in Section 4, whether quantum training can show a practical quantum advantage remains an open question.
* **Discussion of the failure**: We have added a discussion in the revised manuscript (Section 5) discussing when the ''good-enough'' assumption may work or fail.
* **Optimization of the Discussion section**: We have streamlined the content for greater clarity and conciseness, and added a comparison with quantum-inspired related work.

---

### Meta-Review · Area_Chair_Vp4G · 2026-01-07

**Summary:**

Two reviewers were unconvinced on the positive side, with one reject and one marginally below the acceptance threshold. They agreed that this work requires additional effort to meet the acceptance bar of ICLR. Thus, I am inclined not to accept this draft at this stage. Thank you for your effort. It is an interesting work. I hope the input from the reviewers will help you further improve this work.

**Reviewer Concerns:**

This work lack of theoretical guarantee for the proposed hypothesis and needs fine-grained implementation details for the key quantum modules. The experimental results are also limited.

**Reviewer Scores:**

The reviewers' scores reflect the limitations of this work.

---

### Decision · Program_Chairs · 2026-01-26

Reject